# PLANNER-R1: REWARD SHAPING ENABLES EFFICIENT AGENTIC RL WITH SMALLER LLMs

## ABSTRACT

We investigated Agentic RL with large language models on the TRAVELPLAN-NER benchmark. Our approach, PLANNER-R1, achieved a **56.9%** final-pass rate with only 180 training queries, a $2.7\times$ improvement over GPT-5's $21.2\%$ baseline and the strongest agentic result on the public leaderboard. A central finding was that smaller models (8B) were highly responsive to reward shaping: with dense process-level signals, they reached competitive performance while being $3.5\times$ more compute-efficient and $1.5\times$ more memory-efficient than 32B models. Larger models were more robust under sparse rewards but exhibited smaller relative gains from shaping and higher variance across runs. While curriculum learning offered no significant benefit, shaped rewards consistently amplified learning dynamics, making 8B models the most efficient setting for agentic RL. Crucially, these gains did not come at the cost of overfitting: fine-tuned models mostly maintained or exceeded baseline performance on out-of-domain tasks, including MULTI-IF, NAT-URALPLAN, and $\tau$-BENCH. These results establish reward shaping as a decisive lever for scaling agentic RL, highlight the competitive strength of smaller models, and demonstrate that efficiency can be achieved without sacrificing generalization.

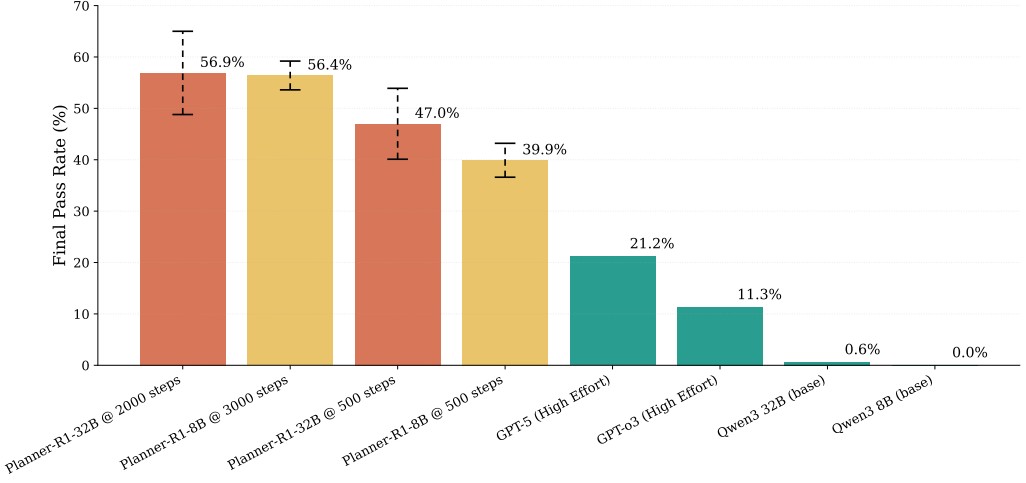

Figure 1: Final-pass rate on the leaderboard test set for tool-use travel planning. Our Planner-R1 models outperformed SOTA LLMs reaching $56.9\%$ average final pass rate.

## 1 INTRODUCTION

Large Language Models (LLMs) have recently posted striking gains in deliberate reasoning and decision making, propelled in part by large-scale reinforcement learning (RL) that trains models to *think before they answer* (OpenAI et al., 2024a; Guo et al., 2025). Beyond language understanding, LLM agents now demonstrate emerging competence in structured reasoning, tool use, and multi-step problem solving across embodied and web environments (Huang et al., 2024; Wang et al.,

2023a; Feng et al., 2025; Zhang et al., 2025b). Yet turning these abilities into *reliable* long-horizon execution under real-world constraints remains challenging: prompting-only agents such as ReAct and Reflexion frequently mis-sequence actions, loop, or hallucinate when tasks demand coordinated tool use and strict constraint satisfaction (Yao et al., 2023b; Shinn et al., 2023).

Planning tasks such as meeting scheduling and multi-day itineraries are demanding: agents must coordinate *heterogeneous tools* (calendars, maps, flights, booking APIs), satisfy *hard, interdependent constraints*, and maintain *global consistency* over long horizons. TRAVELPLANNER makes these difficulties concrete by casting travel itinerary creation as tool-augmented, constraint-driven planning (Xie et al., 2024). The benchmark provides a sandbox with nearly four million records and 1,225 curated intents with reference plans, and evaluates whether an agent can gather evidence via tools and synthesize itineraries that satisfy both explicit user constraints and commonsense feasibility. At release, even strong models struggled—e.g., GPT-4-Turbo with ReAct achieved only a $0.6\%$ *final pass rate* on the 1,000-example test split—underscoring the gap between fluent language modeling and dependable constraint-aware planning (Xie et al., 2024).

To close this gap, researchers have explored different training paradigms. A natural starting point is behavior cloning via supervised fine-tuning (SFT), where a teacher generates "golden" trajectories and a policy maximizes their likelihood, often masking environment observations and tool outputs. While simple and widely used, SFT largely imitates expert behavior and is brittle under distribution shift or suboptimal data. This motivates the search for approaches that directly optimize for end-task success rather than imitation fidelity. RL provides precisely such a mechanism: rewards encode task success, and the policy is updated to increase the likelihood of action sequences that satisfy constraints while suppressing those that fail. Recent work has shown that RL can deliver state-of-the-art gains in model-based reasoning and planning (OpenAI et al., 2024a; Guo et al., 2025), making it a promising direction for tackling long-horizon tool use in TRAVELPLANNER. In addition to model performance, there is growing interest in building efficient agentic systems with smaller models (Belcak et al., 2025). Such models show promising potential for inference and training efficiency, but there remains limited understanding of how agentic RL can best improve their performance without overfitting. Our study addresses this gap by examining how model size, reward shaping, and efficiency interact in agentic RL.

We formulate TRAVELPLANNER as a multi-step, tool-use MDP with constraint-aware planning, where the agent gathers missing facts, reconciles conflicts, and outputs a structured itinerary. Training uses agentic RL with trajectory-level rewards gated by schema validity. Our main focus is the role of *reward density*: we vary feedback from dense, process-level signals to sparse final-pass rewards, and also test a curriculum that transitions between them. All reward variants are *properly shaped*, ensuring they converge to the same optimal policy while revealing how granularity influences learning dynamics. Our contributions are summarized below.

- **SOTA Tool-Use on TravelPlanner** PLANNER-R1-32B achieved a $\mathbf{56.9\%}$ final-pass rate on the official 1,000-query test split, a $2.7\times$ improvement over GPT-5. This is the strongest agentic result on TRAVELPLANNER, demonstrating that RL-tuned models can surpass state-of-the-art proprietary models.[1]

- **Reward shaping dynamics** We find a strong link between reward granularity and policy competence. Smaller models (8B) were especially responsive to shaped, process-level rewards, achieving performance competitive with 32B models while being up to $3.5\times$ more compute-efficient and $1.5\times$ more memory-efficient. Larger models (32B) performed well across reward settings and remained more robust under sparse signals, but exhibited higher variance under dense rewards. In contrast, 8B models depended more heavily on dense shaping. Curriculum learning alone provided no measurable benefit, whereas reward shaping consistently amplified learning dynamics, making the 8B models the most efficient setting for agentic RL.

- **Generalization Beyond Training Domain** Our agents did not overfit to TRAVELPLAN-NER: Planner-R1 models maintained or exceeded baseline performance on out-of-domain tasks including MULTI-IF, NATURALPLAN, and $\tau$-BENCH, provided that excessive domain-specific fine-tuning is avoided. This demonstrates that the efficiency gains from

---

[1]Hao et al. (2025) achieved 93.9% correctness with external SAT/SMT solvers; our focus is on end-to-end agentic planning without such solvers.

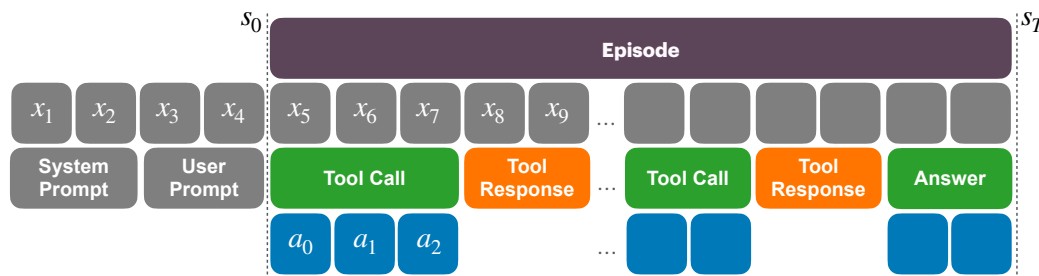

Figure 2: MDP Visualization. $x_i$ represent the $i$th token, while $a_t$ represents the action the agent took at time $t$. Notice that initial prompts and tool responses contain tokens, but they dont increase the time step $t$.

agentic RL come without sacrificing robustness, supporting transfer to diverse planning and tool-use settings.

- **RL Benchmark Formulation** We recast TRAVELPLANNER as a multi-step agentic RL benchmark by leveraging the official sandbox and its seven tools, and we designed verifiable reward functions aligned with the task's success criteria. Policies were trained with VERL(ver, 2024), where our system-level optimizations reduced runtime and memory usage by 20%, enabling efficient large-scale experimentation. (see Appendix A for details)

## 2 PLANNER RL

### 2.1 PROBLEM FORMULATION

We cast tool-augmented planning as a Markov Decision Process (MDP) $\mathcal{M} = (\mathcal{S}, \mathcal{A}, P, r, \gamma)$. Since our MDP is episodic, we set $\gamma = 1$. Each episode is initialized with two textual inputs: a *system prompt $y$*, which defines the agent's role and available tools (see Appendix B.1), and a *user prompt $u$*, which specifies the task goal and user preferences. At each time step $t$, the agent interacts with the environment by emitting a token, alternating between natural language and structured tool invocations, until it decides that a complete plan has been formed. Figure 2 illustrates this process. While our instantiation focuses on the TRAVELPLANNER benchmark (Xie et al., 2024), the formulation is general and extends naturally to other agentic RL tasks. We next describe the individual components of the MDP:

**States.** $s_t \in \mathcal{S}$ denotes the complete history, including the initial system and user prompt, the agent's partial plan, and all tool calls and responses observed up to step $t$, beginning from $s_0 = (y, u)$.

**Actions.** $a_t \in \mathcal{A}$ is the generated token at time $t$. The agent issues *tool calls* through tokens to gather the necessary information and then produces the final plan through a *text action*. Tool calls are realized as seven APIs connected to a sandbox with millions of grounded records: `search_flights`, `search_accommodations`, `search_restaurants`, `search_attractions`, `search_ground_transportation`, `get_cities`, and `calculator`. Each call takes JSON arguments and is wrapped inside `<tool_call>...</tool_call>`, returning a structured JSON object: successful calls yield a list of serialized rows, while failures return an error field.

Compared to the original TRAVELPLANNER, we added the `calculator` API for explicit numeric reasoning and disabled the lightweight semantic memory so that tool responses appear directly in the context. The final text action directly outputs an itinerary enclosed in `<answer>...</answer>`. This design standardizes iterative tool use while keeping the final deliverable unambiguous.

**Transitions.** The environment appends each action to the state; if a tool call is completed, it is executed and the output $o_t$ is added, otherwise $o_t$ is null. The next state is $s_{t+1} = (s_t, a_t, o_{t+1})$, with older context truncated when exceeding the window. A key difference from the original benchmark (Xie et al., 2024) is that we append tokens chronologically to the state, making our transition more generic, as opposed to moving the tool responses to a specific part of the context.

**Reward.** In this domain, success is sparse and binary. A plan receives a reward of one only at termination if it is schema-valid and satisfies both commonsense and user-specified constraints. User queries are designed to ensure that at least one feasible plan exists.

To pass schema validation, the plan must be a valid *JSON array of day-level objects*, each conforming to a fixed schema with fields for `days`, `city`, `transportation`, `attraction`, `accommodation`, `breakfast`, `lunch`, `dinner`. Importantly, `city` and `transportation` are typed objects with required fields (e.g., transportation must specify mode, origin, destination, and duration), rather than free-form strings. The full schema is provided in Appendix B.2.

Constraints fall into two categories. First, there are $N_{\mathrm{cs}}$ *commonsense constraints*, which are not explicitly given to the agent but must nonetheless be satisfied (e.g., transportation segments cannot overlap). Second, there are $N_{\mathrm{hard}}$ *hard constraints*, explicitly specified in the user prompt, such as departure and return dates. Formal definitions and the complete list of constraints are provided in Appendix F and work of Xie et al. (2024).

Our objective is to learn a policy $\pi_\theta(a \mid s)$ that maximizes the expected cumulative reward, which here reduces to optimizing the terminal reward: $\max_\theta \; \mathbb{E}_{\pi_\theta}[r_T]$.

## 2.2 MULTI-STAGE REWARD

Due to the extreme sparsity of the reward function, we shape it using auxiliary metrics defined in the original paper. In particular,

- $r_{\mathrm{schema}} = \mathbb{I}[\text{plan conforms to schema}]$: indicator of schema compliance,
- $r_{\mathrm{cs}}^{\mathrm{micro}} = \frac{S_{\mathrm{cs}}}{N_{\mathrm{cs}}}$: fraction of satisfied commonsense constraints,
- $r_{\mathrm{hard}}^{\mathrm{micro}} = \frac{S_{\mathrm{hard}}}{N_{\mathrm{hard}}}$: fraction of satisfied hard constraints,
- $r_{\mathrm{cs}}^{\mathrm{macro}} = \mathbb{I}\big[r_{\mathrm{cs}}^{\mathrm{micro}} = 1\big]$: indicator that all commonsense constraints pass,
- $r_{\mathrm{hard}}^{\mathrm{macro}} = \mathbb{I}\big[r_{\mathrm{hard}}^{\mathrm{micro}} = 1\big]$: indicator that all hard constraints pass,
- $r_{\mathrm{pass}} = \mathbb{I}[r_{\mathrm{cs}}^{\mathrm{macro}} \wedge r_{\mathrm{hard}}^{\mathrm{macro}}]$: indicator that both commonsense and hard constraints pass.

Here, $\mathbb{I}$ is the indicator function. The micro rewards are necessary to provide partial credit when all constraints are not met , the macro rewards emphasize satisfying entire categories, and $r_{\mathrm{pass}}$ corresponds to the original evaluation metric. The terminal reward in the generic form can then be written as:

$$r = r_{\mathrm{schema}}\Big(\lambda_1 r_{\mathrm{cs}}^{\mathrm{micro}} + \lambda_2 r_{\mathrm{hard}}^{\mathrm{micro}} + \lambda_3 r_{\mathrm{cs}}^{\mathrm{macro}} + \lambda_4 r_{\mathrm{hard}}^{\mathrm{macro}} + \lambda_5 r_{\mathrm{pass}}\Big). \tag{1}$$

By adjusting $\lambda = [\lambda_1, \ldots, \lambda_5]$, we control the reward density. In practice, we consider three stages:

- Stage 1: $\lambda = [1, 1, 1, 1, 1]$ (dense feedback),
- Stage 2: $\lambda = [0, 0, 1, 1, 1]$ (category-level),
- Stage 3: $\lambda = [0, 0, 0, 0, 1]$ (sparse final pass).

This setup defines proper reward shaping: auxiliary terms provide intermediate guidance, while the final-pass reward captures the true objective. Crucially, all of the above weightings preserve the same optimal policy (see proof in Appendix E). Building on this, we define a curriculum that schedules $\lambda$ across training, beginning with dense feedback for partial credit, then shifting to category-level rewards, and finally collapsing to the sparse end reward. Transitions occur at predefined step counts.

## 2.3 OPTIMIZATION

We used GRPO (Shao et al., 2024), a clipped PPO-style objective without KL regularization. For each planning query $u \in \mathcal{D}$, we sample $G$ trajectories $\mathcal{T} = \{\tau_i\}_{i=1}^{G}$ with corresponding Returns $\mathbf{r} = \{r_1, r_2, \cdots, r_G\}$ from the behavior policy $\pi_{\theta_{\mathrm{old}}}$, where $\tau_i = (s_0^i, a_0^i, \ldots, s_{T_i}^i)$. The loss is

$$\mathcal{J}_{\mathrm{GRPO}}(\theta) = \mathbb{E}_{u \sim \mathcal{D},\, \{\tau_i\} \sim \pi_{\theta_{\mathrm{old}}}} \left[ \frac{1}{G} \sum_{i=1}^{G} \frac{1}{T_i} \sum_{t=0}^{T_i - 1} \min\Big( \rho_\theta^{i,t}\, \hat{A}_i,\; \mathrm{clip}(\rho_\theta^{i,t},\, 1 - \epsilon,\, 1 + \epsilon)\, \hat{A}_i \Big) \right], \tag{2}$$

with clipping hyperparameter $\epsilon > 0$. The token-level importance ratio and trajectory-level advantage are defined as

$$\rho_\theta^{i,t} = \frac{\pi_\theta(a_t^i \mid s_t^i, a_{<t}^i)}{\pi_{\theta_{\text{old}}}(a_t^i \mid s_t^i, a_{<t}^i)}, \qquad \hat{A}_i = \frac{r_i - \text{mean}(\mathbf{r})}{\text{std}(\mathbf{r})}.$$

We theoretically proved (proof in Appendix G) that properly shaped rewards like Stage 1 lead to faster convergence than the original sparse Stage 3 reward. This insight provide a fundamental justification for the effectiveness of reward shaping in agentic RL for LLMs. We also proved asymptotic performance equivalence between our sparse and dense rewards under regular assumptions (proof in Appendix H) . This guarantees that the performance of properly shaped rewards is no worse than original sparse reward.

## 3 EMPIRICAL RESULTS

### 3.1 SETUP

**In-Domain**  We fine-tuned Qwen3 8B/32B models across 5 runs with fixed set of seeds on TRAVELPLANNER.[2] Due to GPU memory constraints and the 32K context limit, we conducted preliminary evaluations with "thinking" enabled (qwe, 2025b;a) and found that the additional reasoning quickly saturated the context window, leading to truncated trajectories and degraded performance (see Appendix I). Combined with recent evidence that test-time thinking does not universally improve model quality (Gema et al., 2025; Shojaee* et al., 2025), these observations motivated our decision to disable thinking for all experiments. We named our fine-tuned models Planner-R1. The official 45/180 train–validation split was merged and reshuffled into 180 training and 45 validation queries, while preserving the easy/medium/hard ratio. We evaluated three single-stage reward configurations with 500 steps and a curriculum regime for 8B and 32B models with 100/300/100 and 50/350/100 steps respectively.[3] 8 rollouts were executed in sglang with a standard ReAct-style agent. We capped trajectories at 30 steps, tool responses at $8,192$ tokens, and model outputs at $30,500$ tokens. All runs used two nodes ($16 \times$H200 GPUs). We used learning rate of $10^{-6}$. Full hyperparameters and implementation details are given in Appendix B.3, and decoding and sampling presets are summarized in Appendix B.4.

**Out-of-Domain**  A central concern with task-specific fine-tuning is whether it harms generalization outside the target domain. To probe this, we evaluated our trained models on three complementary suites. All were unseen during training, with evaluation limited to task instructions. (i) NATURAL PLAN (Zheng et al., 2024) (Trip Planning, Meeting Planning, Calendar Scheduling), where tool outputs were provided as context and accuracy was scored by *Exact Match*; we followed the official five-shot prompting protocol. (ii) MULTI-IF (He et al., 2024) (English), a multi-turn instruction-following benchmark derived from IFEval, where the input at turn $t$ concatenated all prior turns ($\leq t - 1$); we reported the mean of turn-wise scores. (iii) $\tau$-BENCH (Yao et al., 2024) (retail, function-calling), which measured goal completion against a simulated backend and policy documents; we reported pass@1.

### 3.2 EVALUATIONS

Table 1 depicts the TRAVELPLANNER results based on Qwen3 (Yang et al., 2025), GPT (OpenAI et al., 2024b; OpenAI, 2025b;a), and our Planner-R1 models using four reward models across six metrics defined in 2. Numbers after $\pm$ indicates 95% confidence intervals.

**Base models showed partial competence but struggled with full constraint satisfaction.**  While stronger base models achieved 99%+ delivery rates and moderate commonsense and hard-constraint coverage, they did not perform well end-to-end. For instance, GPT-5 and GPT-o3 achieved final pass rates of 21.2% and 11.3%, respectively. In contrast, the open-weight Qwen3 series performed substantially worse: the 8B model failed entirely, and the 32B model achieved only 0.6% final pass despite a 41.9% delivery rate. This stark disparity underscored that the challenge lay not in planning

---

[2]We excluded 4B model as preliminary results did not show any trajectory with non-zero return.

[3]Given the strong Stage 3 performance of larger models, we advanced them more quickly from Stage 1.

individual items, but in coordinating tool calls and enforcing all constraints jointly. Prior work (Yao et al., 2023b; Nakano et al., 2021) suggested that prompting alone often underutilized tool feedback, whereas robustness emerged when models interleaved reasoning with actions to query, observe, and update plans. Our findings, as we will see in Section 4, align with this view: base models were able to generate fluent itineraries, but their failures centered on tool sequencing and constraint bookkeeping rather than basic retrieval.

**Agentic RL delivered large gains; smaller models were reward-sensitive.**

RL fine tuning improved both 8B and 32B Qwen3 models. PLANNER-R1-8B with the Stage 1 dense reward reached 39.9%, while PLANNER-R1-32B with Curriculum reward reached 47%. The 8B model was highly sensitive to reward sparsity: Stage 2 and Stage 3 led to 3/5 and 5/5 collapses, consistent with prior evidence that smaller models depend strongly on shaped feedback (Ng et al., 1999; dos Santos et al., 2024; Qian et al., 2025). These collapses are expected. Under sparse Stage 2 and 3 rewards, the 8B model rarely produces correct early trajectories, yielding near-zero advantages and stalling learning within the available compute budget, which also explains the wide confidence intervals for Stage 2 and the 0 performance at Stage 3. Dense process rewards supply graded feedback across micro, macro, and final pass components, allowing steady improvement and effectively acting as a soft curriculum. This implicit progression aligns with ToolRL (Qian et al., 2025), which shows that gradual reward changes outperform hard switches, and explains why explicit curriculum transitions added no measurable benefit in our setup. The 32B model, with a stronger initial policy, can generate partial solutions even under sparse rewards, making it more robust; all reward settings yielded over 42%, with sparsity primarily increasing variance ($\pm 8$ at Stage $1 \rightarrow \pm 14.1$ at Stage 3). Although Curriculum achieved the highest 32B score, the improvement was not statistically significant.

**Smaller models delivered superior GPU efficiency compared to larger ones.** Given the strong performance of Stage 1 training, we extended experiments with high-capacity settings, training the 8B model for 3,000 steps and the 32B model for 2,000 steps. Figure 3 reports results from five independent runs. The left panel, plotted against training steps, shows that both models achieved broadly similar performance trajectories. The right panel, however, plots final pass rate against estimated FLOPs (see Appendix B.6) and reveals a clear efficiency gap. While the 32B model reached 90% of its peak performance (52.3%) at $7.6 \times 10^{20}$ FLOPs, the 8B model achieved the same value at only $2.1 \times 10^{20}$ FLOPs, a $3.5\times$ improvement in efficiency.

Although the 32B model attained a slightly higher peak accuracy (56.9% vs. 56.4%), this difference was not statistically significant and was accompanied by higher variance. We hypothesize that this variance stems from differences in exploration dynamics. TRAVELPLANNER is a challenging environment where essential information is often missing, requiring recovery and alternative solution paths. Smaller models tend to explore less and behave more deterministically, concentrating probability mass on a narrow set of actions. Larger models, by contrast, can consider a broader set of strategies under uncertainty (for example, switching transportation modes or hotel choices), leading to higher variability across rollouts. This exploratory flexibility likely contributes to the increased variance observed in the 32B runs.

A *memory-efficiency* analysis (see Appendix B.5) showed that the 32B model consistently required at least $1.5\times$ more GPU memory than the 8B model, reinforcing that smaller models offer significantly better efficiency for long-context agentic RL when data generation is not the bottleneck.

Table 1: Results on the TRAVELPLANNER test set. For Planner-R1 models, we report mean performance with 95% confidence intervals over five runs at 500 training steps. Stage 1–3 denote runs trained exclusively on one stage for 500 steps each. Curriculum uses three phases: for 8B, 100/300/100 steps; for 32B, 50/350/100 steps across Stages 1–3.

| Method | Delivery Rate (%) | Commonsense | | Hard Constraint | | Final Pass Rate (%) |
|---|---|---|---|---|---|---|
| | | Micro (%) | Macro (%) | Micro (%) | Macro (%) | |
| Qwen3-8B | 0.0 | 0.0 | 0.0 | 0.0 | 0.0 | 0.0 |
| Qwen3-32B | 41.9 | 27.5 | 1.7 | 11.4 | 7.2 | 0.6 |
| GPT-o3 (high) | 99.6 | 74.2 | 14.3 | 57.7 | 48.0 | 11.3 |
| GPT5 (high) | 99.8 | 81.0 | 23.4 | 75.4 | 71.1 | 21.2 |
| Planner-R1-8B | | | | | | |
| Stage1 | 99.5±0.8 | 94.8±1.2 | 69.0±6.9 | 61.0±2.6 | 46.2±2.5 | 39.9±4.3 |
| Stage2 | 99.9±0.2 | 80.6±18.2 | 30.2±51.9 | 63.4±13.8 | 48.6±16.3 | 13.3±23.2 |
| Stage3 | 0.0±0.0 | 0.0±0.0 | 0.0±0.0 | 0.0±0.0 | 0.0±0.0 | 0.0±0.0 |
| Curriculum | 99.7±0.8 | 92.7±3.1 | 57.9±18.6 | 53.9±5.7 | 38.2±4.2 | 27.1±12.6 |
| Planner-R1-32B | | | | | | |
| Stage1 | 99.3±1.6 | 95.2±1.6 | 70.4±13.4 | 74.2±1.4 | 56.4±2.9 | 42.3±8.0 |
| Stage2 | 91.1±0.5 | 87.7±2.2 | 69.1±14.5 | 70.0±5.6 | 55.0±7.6 | 44.1±9.4 |
| Stage3 | 99.4±0.9 | 94.7±2.5 | 71.9±15.2 | 60.8±16.6 | 48.2±15.1 | 44.3±14.1 |
| Curriculum | 99.1±1.7 | 95.9±2.5 | 78.5±7.9 | 72.1±5.0 | 55.1±6.2 | 47.0±6.9 |

**RL fine-tuned models generalized beyond the training domain.** Table 2 shows that RL fine-tuned models performed mostly on par with, and often surpassed, their pretrained counterparts across NATURAL PLAN (Zheng et al., 2024), MULTI-IF (He et al., 2024), and $\tau$-BENCH (Yao et al., 2024). Blue and red indicate significant improvements and degradations, respectively. After 2,000 steps, both models improved on most metrics, and even at 3,000 steps the 8B model outperformed baselines on five of seven metrics with marginal regressions on two metrics. We attribute this robustness to the JSON-gated output structure, which couples semantics with format and reinforces tool-conditioned behaviors, consistent with prior findings that structured generation improves reliability (Oestreich et al., 2025) and supports generalization to unseen schemas (Liu et al., 2019).

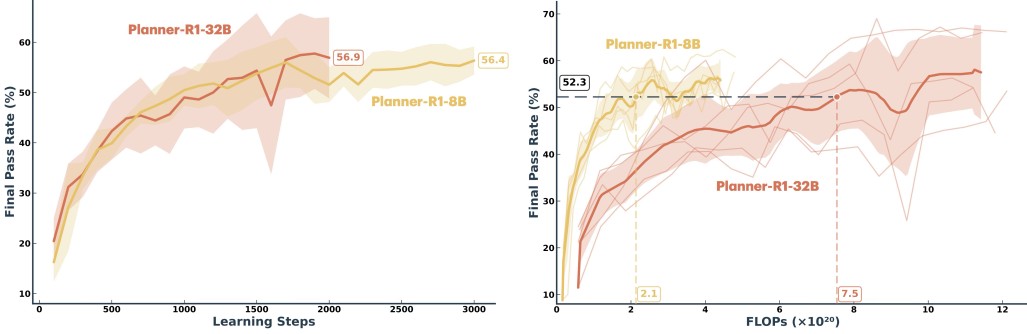

Figure 3: Performance of 8B and 32B Planner-R1 during training based on learning steps (left) and training FLOPS (right). The horizontal dashed line highlights 90% of the maximum average performance of 32B models, while vertical dashed lines show the required FLOPs to reach that performance by both 8B and 32B models.

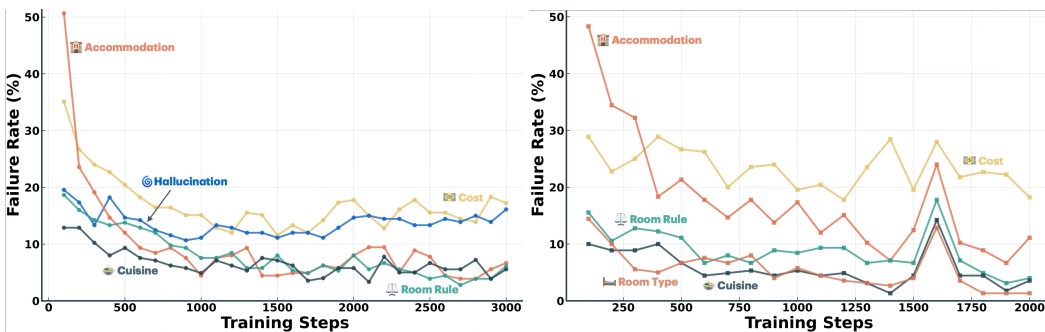

Figure 4: Progression of top 5 failures for 8B (left) and 32B (right) Planner-R1 during training on test set

Table 2: Transferability to external benchmarks without target-domain training (percent). Models are evaluated on NATURAL PLAN, MULTI-IF, and $\tau$-BENCH. Blue = significant improvement over the base model; red = significant degradation from the base model.

| Method (Training Steps) | NATURAL PLAN | | | Multi-IF | | | $\tau$-bench |
|---|---|---|---|---|---|---|---|
| | Trip | Meeting | Calendar | 1st-Turn | 2nd-Turn | 3rd-Turn | Pass@1 |
| Qwen3-8B | 12.9±0.2 | 82.0±0.0 | 22.7±0.3 | 88.9±0.6 | 82.8±0.6 | 75.4±0.5 | 9.5±2.1 |
| Planner-R1-8B (500) | 14.0±0.9 | 83.2±1.1 | 24.3±0.9 | 89.4±0.4 | 83.5±0.7 | 76.9±0.6 | 11.1±0.9 |
| Planner-R1-8B (2000) | 14.0±2.1 | 84.0±0.6 | 23.2±2.1 | 89.8±0.4 | 84.0±0.5 | 77.2±0.4 | 12.1±2.3 |
| Planner-R1-8B (3000) | 10.7±1.8 | 84.5±1.3 | 20.1±2.0 | 89.8±0.1 | 83.9±0.4 | 76.7±0.4 | 15.1±3.1 |
| Qwen3-32B | 11.3±0.0 | 77.0±0.0 | 32.2±0.0 | 89.1±0.3 | 83.1±0.3 | 77.1±0.4 | 28.0±2.2 |
| Planner-R1-32B (500) | 15.7±2.2 | 79.8±1.6 | 33.2±0.5 | 88.7±0.2 | 83.4±0.6 | 77.7±0.6 | 28.7±2.1 |
| Planner-R1-32B (2000) | 19.5±1.2 | 80.2±1.1 | 34.4±1.4 | 89.8±0.3 | 84.1±0.3 | 78.5±0.4 | 33.9±3.8 |

## 4 QUALITATIVE ANALYSIS

To illustrate the effects of RL and model scale, we present a qualitative analysis of Planner-R1 8B and 32B models across training checkpoints, with GPT-5 included as a reference point. We highlight progression in failure modes, tool use, and subreward acquisition for the trained models, and report failure patterns for GPT-5.

**Failure Progression** Figure 4 shows the progression of the top five failure categories for Planner-R1 8B (left) and 32B (right) models during training[4]. For hallucination detection, we verify whether the origin city, destination city, attractions, accommodations, and restaurants are present in the corresponding databases. Both models began with high failure rates, particularly on accommodation and cost constraints. For the 8B model, hallucination and cost remain persistent challenges, while all other failures fall below $10\%$ after 800 steps. For the 32B model, accommodation and cost remain dominant errors, with all other failures dropping below $10\%$ by 600 steps. Notably, the 32B model exhibits substantially fewer hallucinations but struggles more with finding accommodations that qualify, for example when the chosen accommodation has a minimum-night requirement and the planned stay must meet this constraint. Another stark observation is the spike at $1,600$ steps which can be also observed in Figure 3. In our dense–reward setup, the 32B model exhibited noticeably higher variability, with one run dropping from a $44\%$ pass rate to $26\%$ before recovering to $51\%$, which substantially affected the average. As noted earlier, this instability aligns with the larger model's stronger exploratory behavior: the 32B model continues to pursue a broader set of strategies when information is missing, whereas the 8B model behaves more deterministically. This persistent exploratory breadth amplifies variance in the larger model's RL trajectories.

---

[4]Top categories selected based on their AUC during training. For another lens with top 3 failures at each learning step see Figure A.2

**Tool-Use Progression** We observed clear improvements in tool-use behavior as training progressed. Early checkpoints of both the 8B and 32B models exhibited poor sequencing, often looping on repetitive calls (e.g., repeatedly invoking the calculator or restaurant tools), which led to incoherent or incomplete plans. As training progressed, both model failures shifted from syntactic to semantic failures: they returned schema valid plans but often failed to call necessary tools to meet the required constraints. With more training, models could often return valid plans. For further details, see the visualizations of tool-call trajectories in Appendix Figures A.3-A.12.

**Sub-Reward Progression** For both 8B and 32B models, the initial ranking of subrewards from highest to lowest was consistent (see Figure A.1 in Appendix): (1) Schema, (2) Commonsense Micro, (3) Hard Micro, (4) Commonsense Macro, (5) Hard Macro, and (6) Final Pass. As training progressed, success rates increased across all categories, yet this relative ordering remained largely unchanged. This pattern aligned with the $\lambda$ values defined in Section 2.2, reinforcing our intuition about the relative difficulty of these subrewards and underscoring the role of reward shaping in guiding models through progressively harder objectives.

**GPT-5 Behavior** Across multiple scenarios, GPT−5 exhibited several recurring error patterns. These included repetition errors, such as selecting the same restaurant or revisiting a city multiple times, violating commonsense constraints; incomplete plans, such as failing to return to the departing city or omitting key itinerary elements; constraint violations, such as booking fewer than the required minimum number of hotel nights; and hallucinations, including inventing nonexistent hotels or omitting required meals (see Appendix A.4–A.8 for detailed examples).

## 5 RELATED WORK

**Planning** Early *chain-of-thought* prompting showed that writing out intermediate steps boosts LLM performance on complex QA and math (Wei et al., 2022; Kojima et al., 2022). Subsequent variants, most notably self-consistency and structured schemes such as *Least-to-Most* and *Plan-and-Solve*, further reduce errors by decomposing problems and aggregating diverse solution paths (Wang et al., 2023c; Zhou et al., 2023; Wang et al., 2023b). To address the brittleness of linear chains, *search-based* methods recast reasoning as combinatorial exploration with lookahead and backtracking, operating over trees (*Tree of Thoughts*) and graphs (*Graph of Thoughts*) (Yao et al., 2023a; Besta et al., 2024). Multi-agent formulations extend this idea via division of labor: *Chain-of-Agents* partitions long inputs among workers while a manager aggregates their outputs (Chen et al., 2024). Decoupling planning from execution further improves robustness: *Plan-and-Act* pairs a planner with an executor and scales supervision via synthetic trajectories, while *Iterative Programmatic Planning* treats planning as code synthesis (Erdogan et al., 2025; Aravindan et al., 2025). Formal methods offer another angle: Hao et al. (2025) translate planning queries into SAT/SMT specifications solved by external verifiers, achieving rigorous correctness guarantees; in contrast, we keep planning internal to the agent and optimize policies end-to-end with RL. Finally, to reach beyond the context window, recent systems interleave reasoning with targeted search: *Search-o1* triggers agentic retrieval under uncertainty and distills evidence via a Reason-in-Documents step, while *AI-SearchPlanner* trains a lightweight RL planner to trade off query utility and cost, yielding cross-model gains (Li et al., 2025a; Mei et al., 2025). *PilotRL* introduces a planner–executor setup (AdaPlan) optimized with GRPO + VERL, relying on LLM-as-judge rewards and DeepSeek-V3–simulated environments (Lu et al., 2025). Its focus is modular agent optimization; it does not study how reward density interacts with model scale or scaling-related sensitivities.

**Agentic RL** RL is increasingly used to make tool-use strategic and long-horizon: *Search–R1* learns to issue multi-turn web queries during reasoning (Jin et al., 2025), *SkyRL* trains multi-turn agents inside real software environments (Cao et al., 2025), and *ReTool* interleaves Python execution within the reasoning loop under outcome-based rewards (Feng et al., 2025). Complementing these, the Tool-Integrated Reasoning line embeds tools directly into the RL objective: *ToRL* scales tool-integrated RL from base models and reports emergent selective tool invocation with strong math gains (Li et al., 2025b), while *ToolRL* systematically studies reward design for tool selection, showing that shaped rewards with GRPO improve over SFT (Qian et al., 2025). However, its tools and traces are fully synthetic, since LLMs simulate both tool calls and responses, which means it optimizes token imitation rather than performing agentic RL. In contrast, our work conducts true agentic RL with real tools, real observations, and real decision loops. *Biomni* applies end-to-end reinforce-

ment learning, creating rewards and RL environments tailored to biomedicine, scalably training the agent to carry out research tasks more effectively (Huang et al., 2025). In parallel, large-scale RL fine-tuning (*Kimi k1.5*, *DeepSeek–R1*) boosts general reasoning, and *Qwen3* introduces dynamic "thinking" vs. "non-thinking" modes to balance depth and latency (Kimi Team et al., 2025; Guo et al., 2025; Yang et al., 2025). Building on this momentum, *Kimi K2* emphasizes open agentic intelligence with agentic data synthesis and a joint RL stage (Kimi Team, 2025); *GLM-4.5* proposes ARC (Agentic, Reasoning, Coding) foundation models with hybrid thinking/direct modes and RL post-training (GLM-4.5 Team, 2025); and Microsoft's *rStar2–Agent* explores reliable Python tool use with a Resample-on-Correct strategy for agentic RL (Shang et al., 2025). Most closely related, Chen et al. (2025) introduce LOOP, a data- and memory-efficient variant of PPO that enables reinforcement learning for interactive digital agents directly within stateful, multi-domain environments such as AppWorld.

## 6    DISCUSSION

A key finding is that shaped rewards are crucial for smaller models. Under sparse rewards the 8B model failed to learn stable trajectories, but with only 180 training queries, shaped guidance enabled it to reach 32B-level accuracy while delivering **3.5× higher compute efficiency** and **1.5× higher memory efficiency**. Larger models were stable under sparse rewards but gained less from shaping, exhibited higher variance, and incurred higher compute cost, highlighting reward design as a central efficiency lever. These gains at 2,000 steps did not reduce robustness. Fine tuned models matched or exceeded baselines on MULTIIF, NATURALPLAN, and $\tau$ BENCH with no overfitting. At 3,000 steps, the 8B model improved five of seven metrics but regressed on two, indicating the risk of excessive fine tuning. Although we followed leaderboard rules and avoided prompt engineering, both our method and baselines may benefit from prompt refinement.

The remaining 37% gap to the hybrid solver of Hao et al. (2025) is sizable yet tractable. Most remaining errors are valid cost violations (Figure A.2), suggesting difficulty satisfying all hard constraints in a single pass. Recent work on reflect retry reward methods, including REA RL (Deng et al., 2025), SRPO (Wan et al., 2025), and VL Rethinker (Wang et al., 2025), shows that lightweight self verification and self correction can significantly improve constraint satisfaction. Integrating similar mechanisms could close a substantial portion of this gap. A complementary direction is improved sampling. Larger rollout volumes or strategic oversampling with downsampling, as in rStar2 Agent (Shang et al., 2025) and DAPO (Yu et al., 2025), provide richer trajectories and more stable optimization. Although hybrid neuro symbolic solvers remain stronger on constraint heavy tasks, advances in reward shaping, reflection, and sampling can move end to end agentic RL systems much closer to parity, while maintaining the advantage of a single, general solution that transfers across domains rather than a solver engineered for one specific problem.

## 7    LIMITATIONS

We focused on TRAVELPLANNER, a constrained benchmark, and smaller models may not remain competitive on more complex or open-ended tasks. Although 8B models are more FLOP-efficient, larger models still achieve higher peak accuracy and robustness, which can be essential when reliable intermediate feedback is unavailable. Our method also assumes access to dense, process-level rewards, and its effectiveness may drop in settings with only sparse signals. Recent directions help soften this requirement by automating the reward selection methods (Zhang et al., 2025a), using prescribed LLM-as-judge (Lu et al., 2025) or using a structured reward design for tool-selection (Qian et al., 2025). Finally, we used a fixed three-stage curriculum which did not yield advantages yet exploring adaptive or automated scheduling strategies is an important direction for future work. Overall, our results showed that with reliable dense feedback, reward shaping enables smaller models to become strong and compute-efficient alternatives.

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

## A SYSTEM-LEVEL OPTIMIZATIONS

**Overview.** RL training with large-scale LLMs requires co-locating both training and inference engines on the same set of GPUs. This dual demand creates severe memory pressure, often leading to out-of-memory (OOM) errors when switching between training and rollout phases. To address this, we integrated memory management techniques into our RL Pipelines.

**Multi-Stage Awake Memory Management.** In VERL, reinforcement learning (RL) training is conducted in Colocate Mode, where both the training engine (e.g., FSDP) and inference engine (e.g., SGLang) share the same GPU resources. A major bottleneck arises when transferring weights from the training engine to the inference engine: model parameters must be copied from FSDP into SGLang, often causing out-of-memory (OOM) failures under high memory pressure.

To address this, we extended the *Sleep/Awake* mechanism in SGLang and introduced the *Multi-Stage Awake* strategy for fine-grained memory management during rollouts. Instead of a single monolithic resume, memory resumption is divided into multiple stages:

1. Load training model weights into GPU memory.
2. Resume inference model weights at preserved virtual addresses.
3. Synchronize weights between training and rollout engines.
4. Offload training model weights back to CPU.
5. Resume the KV cache region for rollout execution.

This staged approach minimizes memory waste and prevents fragmentation. Our empirical results show that it provides two key benefits:

- **Enables training of larger models**: With the same KV cache ratio, our approach reduces peak GPU memory by **20–23%**, which unblocks stable training of a 32B-parameter model on 8×H200 GPUs even at higher cache ratios (0.8, 0.85, and up to 0.9). Without Multi-Stage Awake, training consistently ran out of memory beyond 0.7.
- **Improves throughput**: For the same model size, our method allows a larger KV cache ratio to be used, directly improving inference throughput. While throughput gains are workload-dependent and not easily comparable across setups, our experiments show that increasing the ratio from 0.7 to 0.9 leads to significant improvements in rollout efficiency.

## B IMPLEMENTATION DETAILS

### B.1 SYSTEM PROMPT

We include the jinja template of our full system prompt used for Planner-R1 during training/evaluation.

```
You are a helpful travel assistant that plans detailed travel
↪   itineraries by calling external functions (tools). You have access
↪   to the following tools and must use them as needed to gather
↪   accurate, up-to-date information.

# Behavior Guidelines
- If a task requires multiple steps or tools, proceed step by step,
↪   calling ONE TOOL per turn.
- Never assume details-always verify all information using tools.
- When you have gathered sufficient information to finalize the plan,
↪   respond with an <answer> block with the final itinerary in valid
↪   JSON format.

# Tool Usage Rules
- Do not repeat the same tool call with identical arguments.
- Always provide complete and correct function arguments.

# Final Plan Format
Once all necessary information is collected, respond with the final
↪   plan:
```
<answer>
  [
    {
      // Day 1 plan following schema
    },
    {
      // Day 2 plan following schema
    },
    // ... additional days
  ]
</answer>
```
**IMPORTANT CONSTRAINTS**
- The <answer> must contain ONLY valid JSON, strictly following the
↪   plan_schema.
- Do not include any explanatory text inside the <answer> block.
- Do not output <answer> until all needed tool calls are completed.

# Final Plan Schema
Each element in the <answer> JSON array should represent a single day
↪   of the trip and follow this schema exactly:
```json
{{ plan_schema }}
```
```

## B.2 PLAN JSON SCHEMA

The final itinerary must be a JSON *array* of per-day objects. Each day object is validated against the schema below. This structured contract doubles as a checklist (ensuring coverage of all required fields) and enables automatic reward gating.

```
{
    "type": "object",
    "required": [
        "days", "city", "transportation", "attraction",
        ↪   "accommodation", "breakfast", "lunch", "dinner"
    ],
    "properties": {
        "days": {
```

```
                  "description": "The day number of the plan starting from
                  ↪   1.",
                  "type": "integer"
              },
          "city": {
              "description": "Can be a city name string if no transfer
              ↪   is needed, or an dict with 'from' and 'to' keys that
              ↪   indicates the origin and destination city.",
              "oneOf": [
                  {"type": "string"},
                  {
                      "type": "object",
                      "required": ["from", "to"],
                      "properties": {
                          "from": {"type": "string"},
                          "to": {"type": "string"}
                      },
                      "additionalProperties": false
                  }
              ]
          },
          "transportation": {
              "description": "Either '-' if no transportation is needed,
              ↪   or an object describing the transportation details.
              ↪   Instead of total cost, use per person price for flight
              ↪   and per vehicle cost for taxi/self-driving as the
              ↪   cost.",
              "oneOf": [
                  {
                      "type": "string",
                      "const": "-"
                  },
                  {
                      "type": "object",
                      "required": ["mode", "from", "to", "duration",
                      ↪   "distance", "cost"],
                      "properties": {
                          "mode": {
                              "type": "string",
                              "enum": ["flight", "taxi",
                              ↪   "self-driving"],
                              "description": "Type of transportation."
                          },
                          "from": {"type": "string", "description":
                          ↪   "Origin city"},
                          "to": {"type": "string", "description":
                          ↪   "Destination city"},
                          "duration": {"type": "string", "description":
                          ↪   "Transportation duration"},
                          "distance": {"type": "string", "description":
                          ↪   "Distance of the trip"},
                          "cost": {"type": "integer", "description":
                          ↪   "Cost of the transportation"},

                          "flight_number": {"type": "string",
                          ↪   "description": "Flight number (for flights
                          ↪   only)"},
                          "departure_time": {"type": "string",
                          ↪   "description": "Flight departure time"},
                          "arrival_time": {"type": "string",
                          ↪   "description": "Flight arrival time"}
                      },
                      "additionalProperties": false
                  }
```

```
                    ]
            },
            "attraction": {
                "description": "A list of attraction names planned for the
                ↪  day, or '-' if no attractions are planned.",
                "oneOf": [
                    {"type": "string", "const": "-"},
                    {
                        "type": "array",
                        "items": {"type": "string"},
                        "minItems": 1
                    }
                ]
            },
            "accommodation": {
                "description": "The name of the accommodation for today.
                ↪  '-' if no accommodation is needed.",
                "type": "string"
            },
            "breakfast": {
                "description": "The name of the breakfast restaurant for
                ↪  today. '-' if no breakfast is planned.",
                "type": "string"
            },
            "lunch": {
                "description": "The name of the lunch restaurant for
                ↪  today. '-' if no lunch is planned.",
                "type": "string"
            },
            "dinner": {
                "description": "The name of the dinner restaurant for
                ↪  today. '-' if no dinner is planned.",
                "type": "string"
            }
        },
        "additionalProperties": false
}
```

### B.3  TRAINING, VALIDATION, AND EVALUATION SETUP

**RL framework and resources.**   We train with VERL using **GRPO** on **2 nodes** with **8 GPUs/node** (16 H200 GPUs total). Rollouts use **sglang** with a multi-turn, tool-augmented agent (ReAct-style).

**Training configuration.**   Stage 1/2/3 share the same VERL configuration; only the reward weights differ by stage (see Sec. 2). File paths are pseudonymized for readability.

```
# verl + GRPO. Stage-agnostic; change reward weights per stage.
actor_rollout_ref:
  actor:
    strategy: fsdp
    ppo_mini_batch_size: 8
    ppo_micro_batch_size: null
    ppo_micro_batch_size_per_gpu: 1
    use_dynamic_bsz: false
    ppo_max_token_len_per_gpu: 16384
    clip_ratio: 0.2
    clip_ratio_low: 0.2
    clip_ratio_high: 0.2
    policy_loss:
      loss_mode: vanilla
```

```
      clip_cov_ratio: 0.0002
      clip_cov_lb: 1.0
      clip_cov_ub: 5.0
      kl_cov_ratio: 0.0002
      ppo_kl_coef: 0.1
    clip_ratio_c: 3.0
    loss_agg_mode: token-mean
    entropy_coeff: 0
    use_kl_loss: false
    use_torch_compile: true
    kl_loss_coef: 0.001
    kl_loss_type: low_var_kl
    ppo_epochs: 1
    shuffle: false
    optim:
      lr: 1.0e-06
      lr_warmup_steps_ratio: 0.0
      total_training_steps: -1
      weight_decay: 0.01
      lr_warmup_steps: -1
      min_lr_ratio: 0.0
      num_cycles: 0.5
      warmup_style: constant
    grad_clip: 1.0
    ulysses_sequence_parallel_size: 1
    entropy_from_logits_with_chunking: false
    entropy_checkpointing: false
    fsdp_config:
      wrap_policy:
        min_num_params: 0
      param_offload: true
      optimizer_offload: true
      offload_policy: false
      reshard_after_forward: true
      fsdp_size: -1
      forward_prefetch: false
  rollout:
    name: sglang
    mode: async
    temperature: 1.0
    top_k: -1
    top_p: 1
    prompt_length: 2268
    response_length: 30500
    dtype: bfloat16
    gpu_memory_utilization: 0.6
    ignore_eos: false
    enforce_eager: true
    free_cache_engine: true
    tensor_model_parallel_size: 4
    max_num_batched_tokens: 8192
    max_model_len: null
    max_num_seqs: 1024
    log_prob_micro_batch_size: null
    log_prob_micro_batch_size_per_gpu: 32
    log_prob_use_dynamic_bsz: false
    log_prob_max_token_len_per_gpu: 16384
    disable_log_stats: true
    do_sample: true
    n: 8
    multi_stage_wake_up: false
    val_kwargs:
      top_k: -1
      top_p: 1.0
```

```
      temperature: 0
      n: 1
      do_sample: false
    multi_turn:
      enable: true
      max_assistant_turns: 30
      tool_config_path: ${PROJ_ROOT}/config/tool_config.yaml
      max_user_turns: 30
      max_parallel_calls: 1
      max_tool_response_length: 8192
      tool_response_truncate_side: right
      interaction_config_path: null
      completion_callback: null
      use_inference_chat_template: false
      tokenization_sanity_check_mode: strict
      format: hermes
    calculate_log_probs: false
    agent:
      num_workers: 8
      agent_loop_config_path: ${PROJ_ROOT}/config/agent_loops.yaml
      custom_async_server:
        path: null
        name: null
    update_weights_bucket_megabytes: 512
    enable_chunked_prefill: true
    load_format: dummy_dtensor
    layered_summon: false
    enable_thinking: false
  hybrid_engine: true
  model:
    path: Qwen/Qwen3-{8B|32B}  # base model
    custom_chat_template: null
    use_shm: false
    external_lib: null
    override_config: {}
    enable_gradient_checkpointing: true
    enable_activation_offload: false
    use_remove_padding: true
    target_modules: all-linear
    exclude_modules: null
    use_liger: false
    use_fused_kernels: false
    fused_kernel_options:
      impl_backend: torch
    trust_remote_code: false
trainer:
  balance_batch: true
  total_epochs: 300
  total_training_steps: 3000
  profile_steps: null
  logger:
    - mlflow
  log_val_generations: 0
  rollout_data_dir: null
  nnodes: 2
  n_gpus_per_node: 8
  save_freq: 100
  esi_redundant_time: 0
  resume_mode: auto
  val_before_train: true
  val_only: false
  test_freq: 50
  critic_warmup: 0
  default_hdfs_dir: null
```

```
  del_local_ckpt_after_load: false
  max_actor_ckpt_to_keep: null
  max_critic_ckpt_to_keep: null
  ray_wait_register_center_timeout: 300
  device: cuda
  use_legacy_worker_impl: auto
data:
  tokenizer: null
  use_shm: false
  train_files: ${PROJ_ROOT}/data/train.parquet
  val_files: ${PROJ_ROOT}/data/test.parquet
  prompt_key: prompt
  reward_fn_key: data_source
  max_prompt_length: 2268
  max_response_length: 30500
  train_batch_size: 16
  val_batch_size: 64
  return_raw_input_ids: false
  return_raw_chat: true
  return_full_prompt: false
  shuffle: true
  dataloader_num_workers: 8
  validation_shuffle: false
  filter_overlong_prompts: true
  filter_overlong_prompts_workers: 1
  truncation: error
  image_key: images
  video_key: videos
  trust_remote_code: false
custom_reward_function:
  path: ${PROJ_ROOT}/rewards_v3.py
  name: compute_score
algorithm:
  gamma: 1.0
  lam: 1.0
  adv_estimator: grpo
  norm_adv_by_std_in_grpo: true
  use_kl_in_reward: false
  kl_penalty: kl
  kl_ctrl:
    type: fixed
    kl_coef: 0.001
    horizon: 10000
    target_kl: 0.1
  use_pf_ppo: false
  pf_ppo:
    reweight_method: pow
    weight_pow: 2.0
```

Listing 1: VERL/GRPO configuration (paths pseudonymized). Stage 2/3 reuse this config with stage-specific reward weights.

**Evaluation protocol.** We evaluate on the TRAVELPLANNER *official test set* by reusing the VERL *validation* pipeline to keep decoding/sampling consistent with validation:

1. Point the VERL validation loader to the official test split (same sampler settings as validation).

2. Run validation to **dump trajectories** locally (tool calls, responses, final answers) as JSONL.

3. **Post-process** each final answer: validate against the plan schema (App. B.2), enforce JSON-gated output, and **convert** to the leaderboard's submission format.

4. **Upload** the converted file to the TRAVELPLANNER leaderboard; report Delivery, micro/macro commonsense, micro/macro hard, and Final.

### B.4 DECODING AND SAMPLING SETTINGS

We standardize decoding across training and evaluation to isolate the effect of learning. Table 3 summarizes the presets we use for different contexts; "Common runtime limits" apply to all scenarios unless noted. For Validation/Test we reuse VERL's validation path on the official TP test split (Sec. B.3).

**Common runtime limits.**

- **Max response tokens:** 30,500                                     (`response_length`)
- **Max tool response tokens:** 8,192                          (`max_tool_response_length`)
- **Agent turns cap:** 30 assistant turns; 30 tool turns
- **Tool-call cap:** 30 calls

Table 3: Decoding presets by context.

| Context | do_sample | Temp | Top-$p$ | Top-$k$ | $n$ |
|---|---|---|---|---|---|
| Training | true | 1.0 | 1.0 | $-1$ | 8 |
| Validation / Test | false | 0.0 | 1.0 | $-1$ | 1 |
| NaturalPlan | true | 1.0 | 1.0 | $-1$ | 1 |
| Multi-IF | false | 0.7 | 0.8 | 20 | 1 |
| $\tau$-bench | false | 0.6 | 0.95 | 20 | 1 |

### B.5 GPU MEMORY FOOTPRINT AND PRACTICAL EFFICIENCY

In the same 2-node/16-GPU training setup, PLANNER-R1-8B used approximately $\sim$60 GB of GPU memory per device, whereas PLANNER-R1-32B required $\geq$90 GB (1.5$\times$ more) per device. This difference has practical consequences: the 8B configuration ran comfortably on H100s, while the 32B configuration necessitated higher-memory accelerators (e.g., H200). The gap is especially relevant for agentic RL, where multi-turn interactions and tool feedback produce long contexts and large key–value (KV) caches during rollouts, amplifying the memory pressure beyond the update phase.

### B.6 ESTIMATING TRAINING FLOPs FROM VERL'S MFU

**MFU in VERL (what it is).** VERL reports a *model FLOPs utilization* (MFU): the fraction of the cluster's "promised" peak compute achieved during *policy updates*. Internally it is computed per update as

$$\text{MFU} = \frac{f_{\text{ach}}\, E}{f_{\text{peak}}\, W}, \qquad f_{\text{ach}} = \frac{\text{FLOPs}_{\text{update}}}{t_{\text{actor}}},$$

where $E$ is the number of GRPO epochs per batch, $W$ is the number of GPUs (world size), $f_{\text{peak}}$ is the *promised FLOPs rate per GPU* used by VERL in its MFU denominator, $t_{\text{actor}}$ is the time spent in the parameter-update step, and $\text{FLOPs}_{\text{update}}$ is the per-step FLOPs consumed by that update. The in-tree FLOPs counter aggregates *forward + backward* over all layers/tokens.

**Reconstruction used in this paper.** Solving for $\text{FLOPs}_{\text{update}}$ gives

$$\boxed{\text{FLOPs}_{\text{update}} = \text{MFU} \times f_{\text{peak}} \times W \times \frac{t_{\text{actor}}}{E}}$$

In our runs $E{=}1$ and $W{=}16$. We set $f_{\text{peak}}{=}9.89 \times 10^{14}$ FLOPs/s per GPU—the same constant VERL uses for MFU—so the reconstruction matches its calculation.

**Practical proxy for $t_{\text{actor}}$.** VERL does not log $t_{\text{actor}}$ each step, but it logs `update_policy_time` ($t_{\text{policy}}$), which equals the actor update plus brief offload/reload bookkeeping. Because the parameter update dominates, we use

$$t_{\text{actor}} \approx t_{\text{policy}} \quad \Rightarrow \quad \text{FLOPs}_{\text{update}} \approx \text{MFU} \times f_{\text{peak}} \times W \times t_{\text{policy}}.$$

This yields a *slight upper bound* (since $t_{\text{policy}} \geq t_{\text{actor}}$); spot checks in our regime found the gap within $\sim$3%.

**From per-step to cumulative FLOPs.** We compute $\text{FLOPs}_{\text{update}}$ per step from MFU and $t_{\text{policy}}$, then sum over steps:

$$\text{FLOPs}_{1:T} \;=\; \sum_{t=1}^{T} \text{MFU}_t \; f_{\text{peak}} \; W \; t_{\text{policy},t} \;.$$

These cumulative totals back the FLOPs–accuracy curves in Sec. 3.2.

**Scope: what is counted and what is not.** Our accounting *includes only* the parameter-update compute (forward + backward). It *excludes* (i) the rollout engine's generation compute and (ii) the *reference log-prob* pass (both of which VERL does not report MFU/FLOPs for). Under our settings (long responses, multi-sample trajectories), rollout consists of many forward passes, while the update consists of forward+backward over similar tokens; thus their compute is typically of the same *order of magnitude*, but exact ratios depend on response length, batching, and $n$ (number of sampled trajectories). A precise FLOPs tally for rollout and reference log-prob computation is left to future work.

## C  FURTHER ANALYSIS

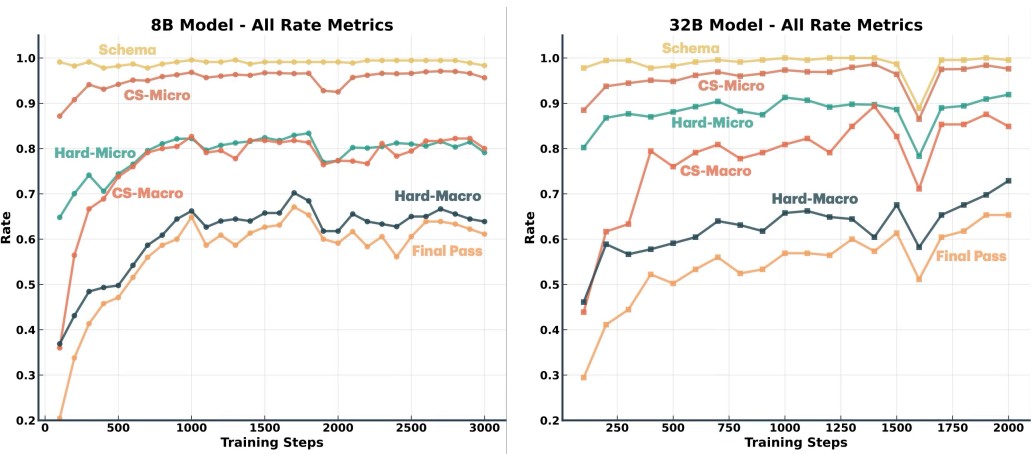

Figure A.1: Progression of six sub rewards for 8B and 32B Planner-R1 during training on validation set

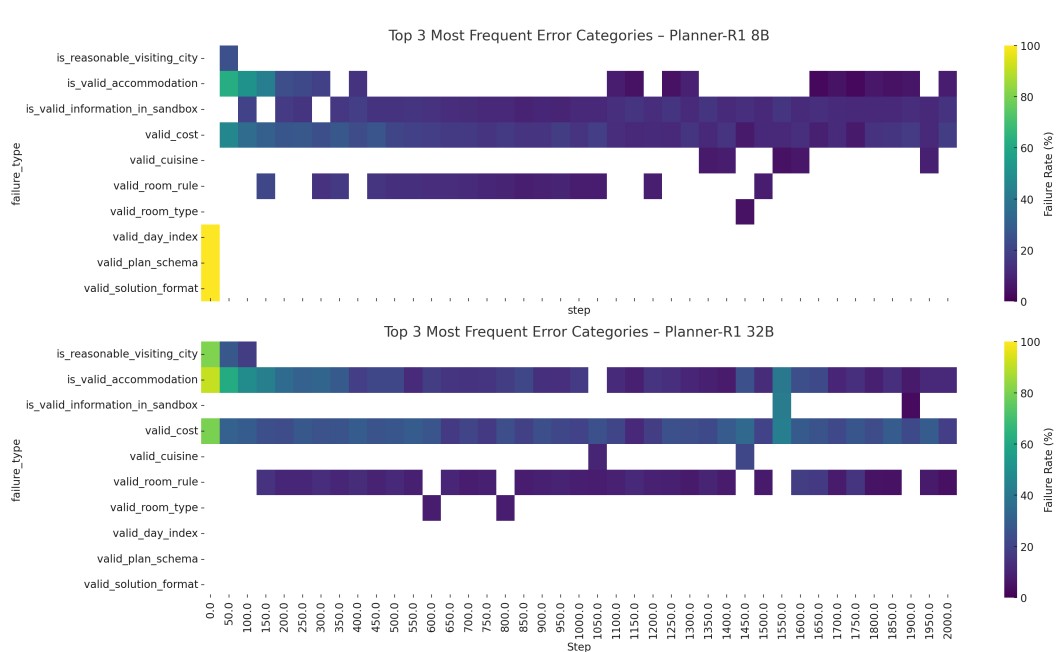

Figure A.2: Top 3 Most Frequent Error Categories for the Planner-R1 models. The upper heatmap shows the 8B model and the lower heatmap shows the 32B model. Both plots visualize, at each training step across five runs (up to 2000), the three most frequent failure categories and their relative rates. Rows are aligned so categories match across models; blank cells indicate that a category did not appear in the top three for that model at that step.

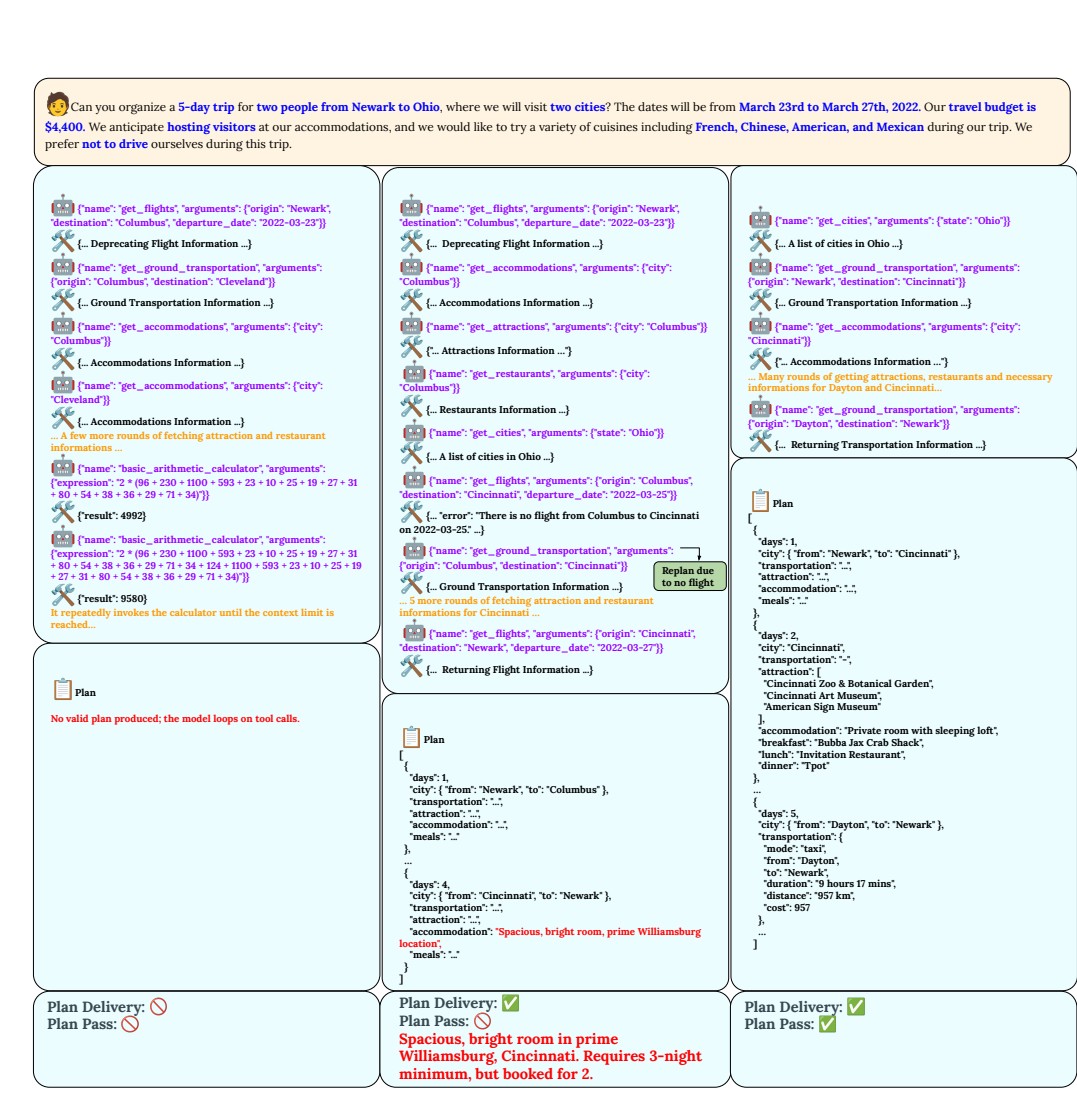

Figure A.3: Model tool-call trajectories across checkpoints. The base model (left) loops on repetitive tool calls once the context is saturated. After 100 training steps (middle), the model produces a coherent travel plan but fails to satisfy all constraints. By 500 steps with the 32B Planner-R1 checkpoint (right), the model successfully generates a valid plan that meets all requirements.

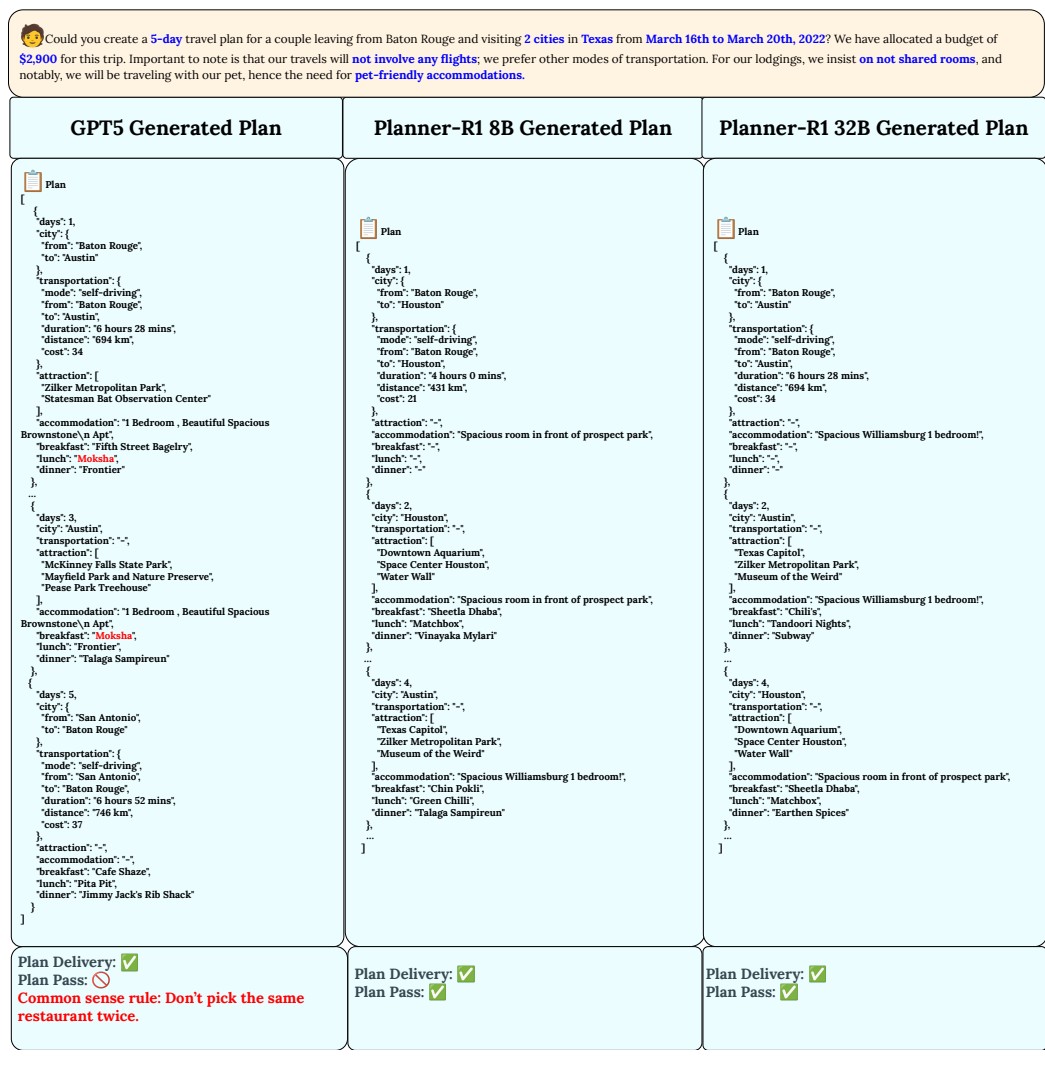

Figure A.4: The GPT-5 model (left) failed to avoid selecting the same restaurant twice, thus violating the common-sense rule. In contrast, the Planner-R1 8B model (middle) and Planner-R1 32B model (right) both generated plans that satisfied all requirements.

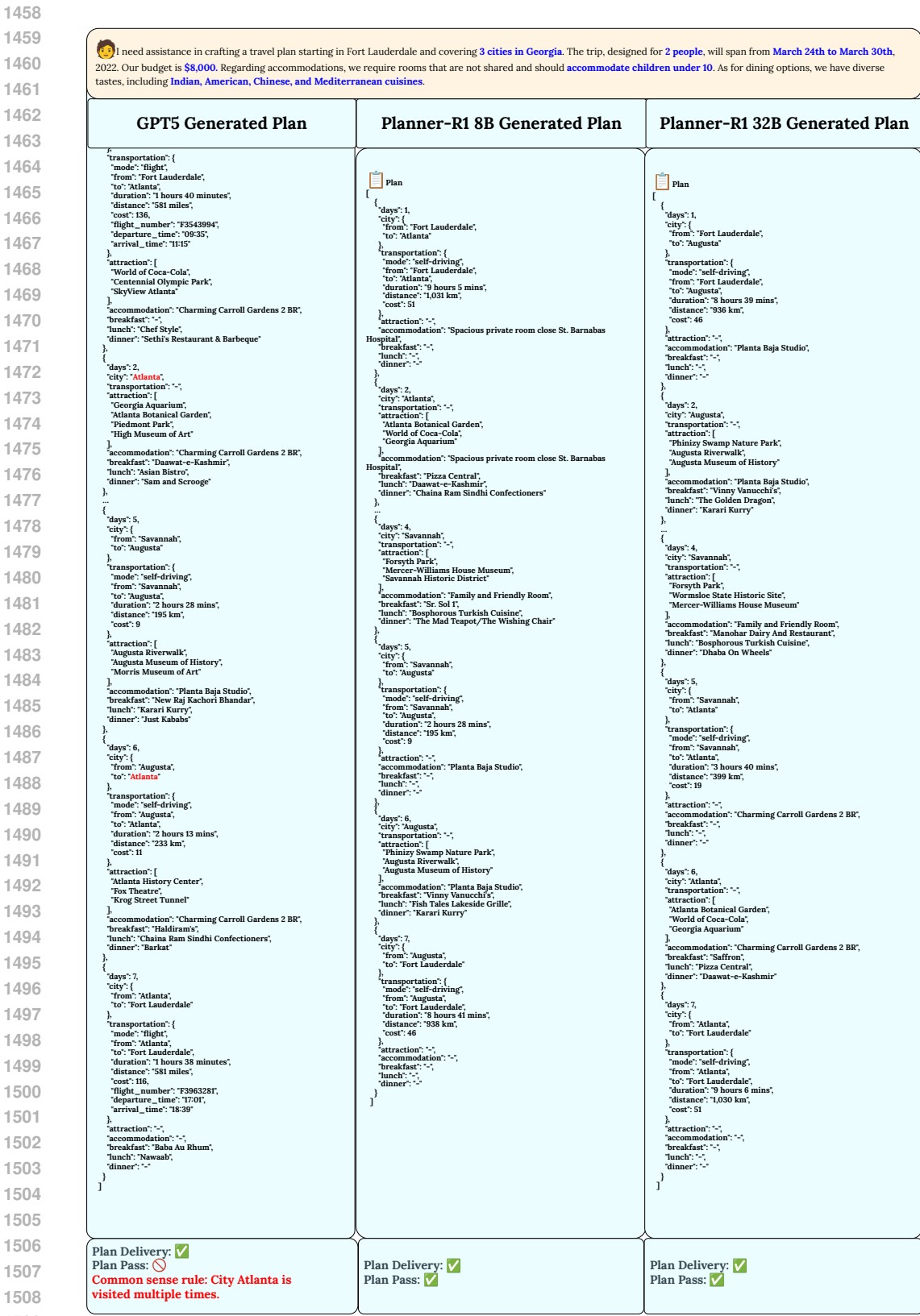

Figure A.5: The GPT-5 model (left) selected the same restaurant twice on different dates, whereas the Planner-R1 8B (middle) and Planner-R1 32B (right) models produced plans that satisfied all requirements.

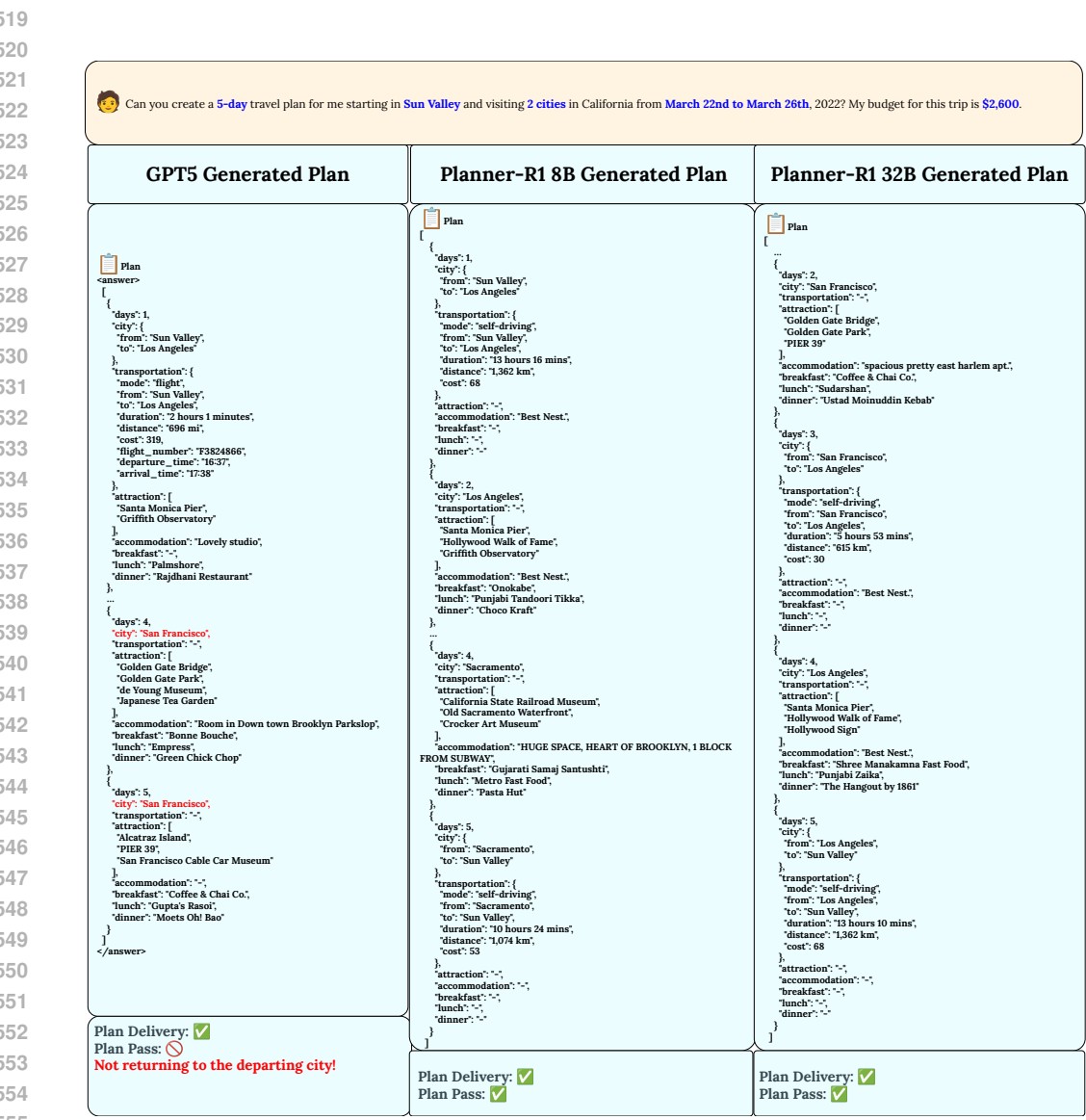

Figure A.6: The GPT-5 model (left) failed to return to the departing city, whereas the Planner-R1 8B (middle) and Planner-R1 32B (right) models produced plans that satisfied all requirements.

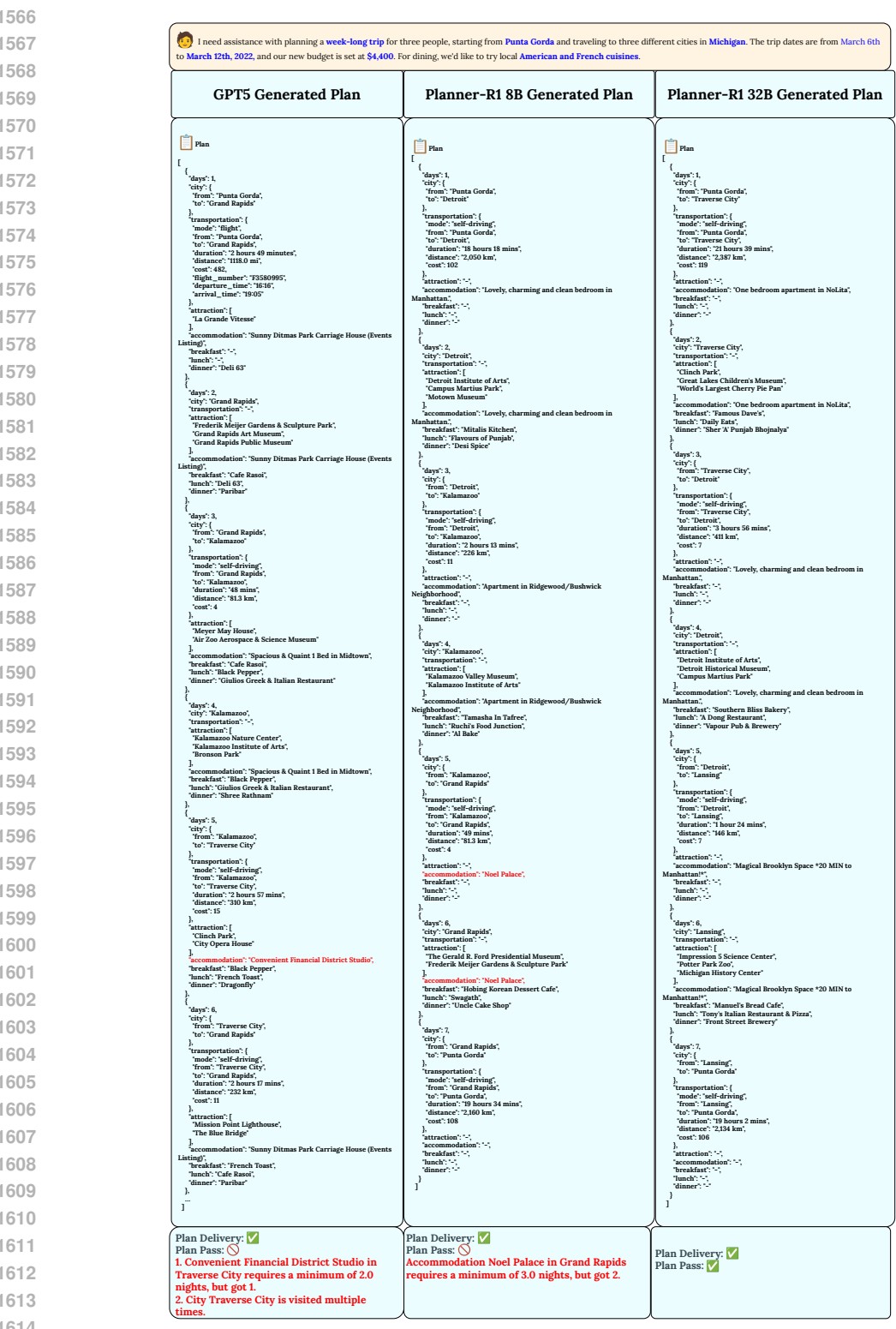

Figure A.7: The GPT-5 model (left) violated the hotel booking rule by reserving only one night instead of the required minimum of two, and also erred by visiting Traverse City multiple times, failing the common-sense requirement. The Planner-R1 8B model (middle) likewise failed the accommodation requirement, while only the Planner-R1 32B model (right) satisfied all requirements.

> 🧑 Could you create a travel plan for a **party of 8** departing from **Fayetteville** and heading to New York for 5 days, which will cover **2 cities** between **March 25th and March 29th, 2022**? We have a maximum budget of **$6,900**. We are particularly interested in experiencing **American and Mexican cuisines** during our stay.

| GPT5 Generated Plan | Planner-R1 8B Generated Plan | Planner-R1 32B Generated Plan |
|---|---|---|

📋 **Plan** (GPT5)

```json
[
  {
    "days": 1,
    "city": {
      "from": "Fayetteville",
      "to": "New York"
    },
    "transportation": {
      "mode": "flight",
      "from": "Fayetteville",
      "to": "New York",
      "duration": "3 hours 8 minutes",
      "distance": "1147.0 mi",
      "cost": 260,
      "flight_number": "F4057738",
      "departure_time": "07:15",
      "arrival_time": "11:23"
    },
    "attraction": [
      "Times Square",
      "Top of The Rock",
      "Central Park"
    ],
    "accommodation": "Magical rowhouse and garden in Williamsburg (3 units)",
    "breakfast": "-",
    "lunch": "Gurgaon Hights",
    "dinner": "G Dot"
  },
  {
    "days": 2,
    "city": "New York",
    "transportation": "-",
    "attraction": [
      "Statue of Liberty",
      "9/11 Memorial & Museum",
      "Brooklyn Bridge",
      "The High Line"
    ],
    "accommodation": "Magical rowhouse and garden in Williamsburg (3 units)",
    "breakfast": "-",
    "lunch": "Amchur",
    "dinner": "Gurgaon Hights"
  },
  {
    "days": 3,
    "city": {
      "from": "New York",
      "to": "Newark"
    },
    "transportation": {
      "mode": "taxi",
      "from": "New York",
      "to": "Newark",
      "duration": "29 mins",
      "distance": "16.3 km",
      "cost": 16
    },
    "attraction": [
      "The Newark Museum of Art",
      "Branch Brook Park",
      "Military Park"
    ],
    "accommodation": "Magical rowhouse and garden in Williamsburg (3 units)",
    "breakfast": "-",
    "lunch": "Anaicha's Food Joint",
    "dinner": "Mirage Restro Bar"
  },
  {
    "days": 4,
    "city": "New York",
    "transportation": "-",
    "attraction": [
      "Empire State Building",
      "Rockefeller Center",
      "Radio City Music Hall"
    ],
    "accommodation": "Magical rowhouse and garden in Williamsburg (3 units)",
    "breakfast": "-",
    "lunch": "G Dot",
    "dinner": "Amchur"
  },
  {
    "days": 5,
    "city": "New York",
    "transportation": "-",
    "attraction": [
      "One World Observatory",
      "The Battery",
      "Flatiron Building"
    ],
    "accommodation": "-",
    "breakfast": "-",
    "lunch": "Gurgaon Hights",
    "dinner": "G Dot"
  }
]
```

📋 **Plan** (Planner-R1 8B)

```json
[
  {
    "days": 1,
    "city": {
      "from": "Fayetteville",
      "to": "New York"
    },
    "transportation": {
      "mode": "self-driving",
      "from": "Fayetteville",
      "to": "New York",
      "duration": "8 hours 19 mins",
      "distance": "873 km",
      "cost": 43
    },
    "attraction": "-",
    "accommodation": "Modern Brooklyn oasis (PRIVATE ROOM)",
    "breakfast": "-",
    "lunch": "-",
    "dinner": "-"
  },
  {
    "days": 2,
    "city": "New York",
    "transportation": "-",
    "attraction": [
      "Top of The Rock",
      "Rockefeller Center",
      "Empire State Building",
      "The High Line"
    ],
    "accommodation": "Modern Brooklyn oasis (PRIVATE ROOM)",
    "breakfast": "Gurgaon Hights",
    "lunch": "G Dot",
    "dinner": "Aryan's Rajasthani Pyaz Ki Kachori"
  },
  {
    "days": 3,
    "city": {
      "from": "New York",
      "to": "Boston"
    },
    "transportation": {
      "mode": "self-driving",
      "from": "New York",
      "to": "Boston",
      "duration": "3 hours 41 mins",
      "distance": "346 km",
      "cost": 17
    },
    "attraction": "-",
    "accommodation": "Sunlight + Space on Eastern Parkway",
    "breakfast": "-",
    "lunch": "-",
    "dinner": "-"
  },
  {
    "days": 4,
    "city": "Boston",
    "transportation": "-",
    "attraction": [
      "Public Garden",
      "Museum of Science",
      "New England Aquarium"
    ],
    "accommodation": "Sunlight + Space on Eastern Parkway",
    "breakfast": "Irish Democrat",
    "lunch": "Apni Rasoi",
    "dinner": "Resunga Food Corner"
  },
  {
    "days": 5,
    "city": {
      "from": "Boston",
      "to": "Fayetteville"
    },
    "transportation": {
      "mode": "self-driving",
      "from": "Boston",
      "to": "Fayetteville",
      "duration": "11 hours 42 mins",
      "distance": "1,226 km",
      "cost": 61
    },
    "attraction": "-",
    "accommodation": "-",
    "breakfast": "-",
    "lunch": "-",
    "dinner": "-"
  }
]
```

📋 **Plan** (Planner-R1 32B)

```json
[
  {
    "days": 1,
    "city": {
      "from": "Fayetteville",
      "to": "Buffalo"
    },
    "transportation": {
      "mode": "self-driving",
      "from": "Fayetteville",
      "to": "Buffalo",
      "duration": "11 hours 11 mins",
      "distance": "1,169 km",
      "cost": 58
    },
    "attraction": "-",
    "accommodation": "Williamsburg Gem: Sleep up to 5",
    "breakfast": "-",
    "lunch": "-",
    "dinner": "-"
  },
  {
    "days": 2,
    "city": "Buffalo",
    "transportation": "-",
    "attraction": [
      "The Buffalo Zoo",
      "Buffalo and Erie County Botanical Gardens",
      "Buffalo AKG Art Museum"
    ],
    "accommodation": "Williamsburg Gem: Sleep up to 5",
    "breakfast": "Red Mango",
    "lunch": "Tibby's New Orleans Kitchen",
    "dinner": "Shokitini"
  },
  {
    "days": 3,
    "city": {
      "from": "Buffalo",
      "to": "New York"
    },
    "transportation": {
      "mode": "self-driving",
      "from": "Buffalo",
      "to": "New York",
      "duration": "6 hours 14 mins",
      "distance": "600 km",
      "cost": 30
    },
    "attraction": "-",
    "accommodation": "A Contemporary Homelike Stay in the Best of BK",
    "breakfast": "-",
    "lunch": "-",
    "dinner": "-"
  },
  {
    "days": 4,
    "city": "New York",
    "transportation": "-",
    "attraction": [
      "Top of The Rock",
      "One World Observatory",
      "SUMMIT One Vanderbilt"
    ],
    "accommodation": "A Contemporary Homelike Stay in the Best of BK",
    "breakfast": "Gurgaon Hights",
    "lunch": "G Dot",
    "dinner": "Rambhog"
  },
  {
    "days": 5,
    "city": {
      "from": "New York",
      "to": "Fayetteville"
    },
    "transportation": {
      "mode": "self-driving",
      "from": "New York",
      "to": "Fayetteville",
      "duration": "8 hours 21 mins",
      "distance": "881 km",
      "cost": 44
    },
    "attraction": "-",
    "accommodation": "-",
    "breakfast": "-",
    "lunch": "-",
    "dinner": "-"
  }
]
```

**GPT5:**
Plan Delivery: ✅
Plan Pass: 🚫
1. Missing meals on day 2.
2. Accommodation Magical rowhouse and garden is not presented in database

**Planner-R1 8B:**
Plan Delivery: ✅
Plan Pass: ✅

**Planner-R1 32B:**
Plan Delivery: ✅
Plan Pass: ✅

Figure A.8: In this query, the user plans a trip for a party of 8. The GPT-5 model (left) missed meals on Day 2 and hallucinated non-existent hotel names. The Planner-R1 8B model (middle) generated a plan that exceeded the $6,900 budget, while only the Planner-R1 32B model (right) satisfied all requirements.

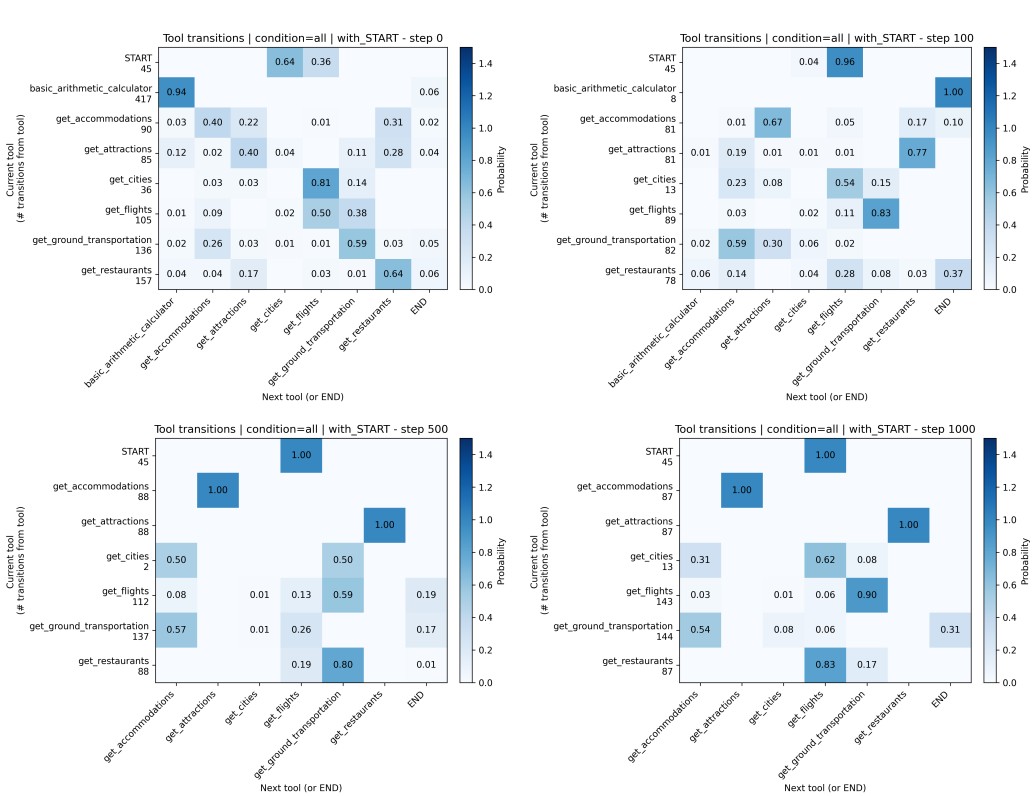

Figure A.9: Policy visualization for 8B model across 45 trajectories based on previous (y-axis) and next (x-axis) tool calls across various steps of learning: $\{0, 100, 500, 1000\}$. As learning progresses, the policy becomes more deterministic.

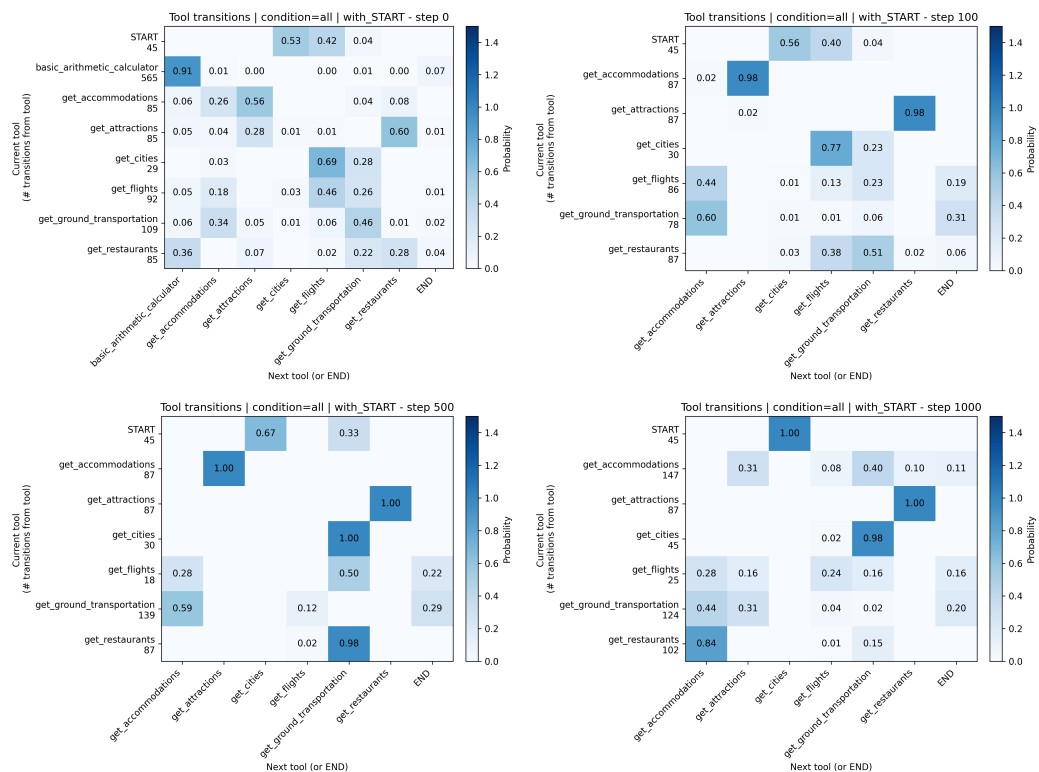

Figure A.10: Policy visualization 32B model across 45 trajectories based on previous (y-axis) and next (x-axis) tool calls across various steps of learning: $\{0, 100, 500, 1000\}$. As learning progresses, the policy becomes more deterministic.

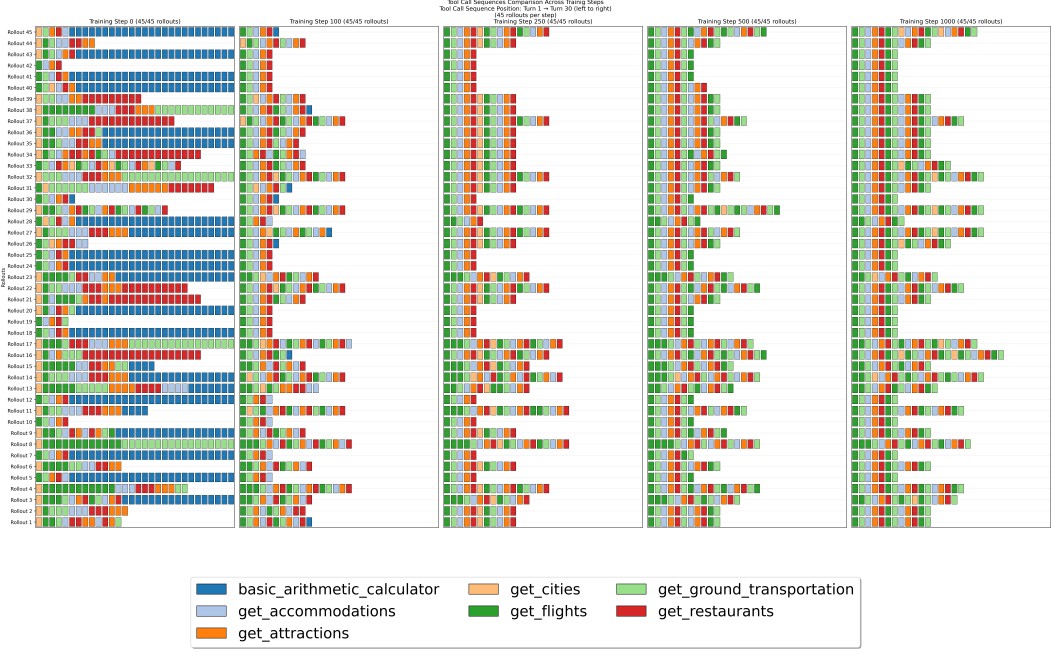

Figure A.11: Tool call sequence behavior as 8B training progresses. The base model (leftmost) repeatedly invoked the calculator and restaurant tools until reaching the rollout cap (30 turns), exhibiting poor tool-use behavior as context grew. With longer training, the model developed more consistent and structured patterns for tool calls.

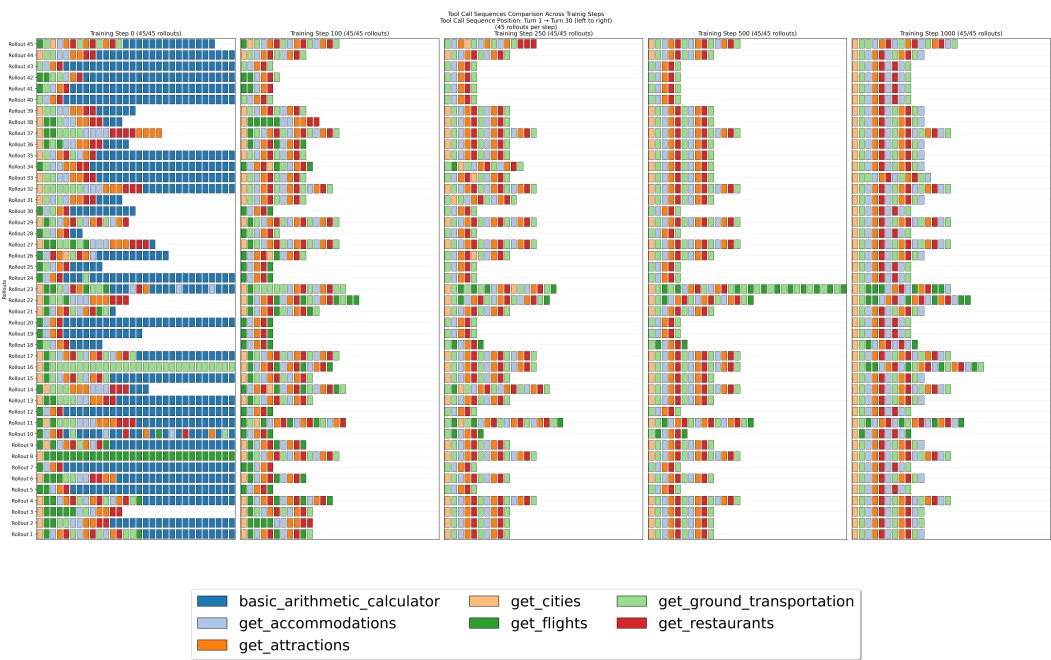

Figure A.12: Tool call sequence behavior as 32B training progresses. The base model (leftmost) repeatedly invoked the calculator until reaching the rollout cap (30 turns), exhibiting poor tool-use behavior as context grew, similar to the 8B model. With longer training, the model developed more consistent and structured patterns for tool calls. In particular, it learned to invoke get cities early to check available cities within states before searching for tickets and attractions. We also observed that the model made fewer get flights calls across queries, instead preferring to select more grounded transportation options.

## D  LLM USAGE DISCLOSURE

We used large language models (LLMs) *only* as general-purpose writing assistants for surface-level language edits (grammar, phrasing, and clarity), LaTeX formatting suggestions (e.g., table/figure spacing, caption wording), and copyediting (e.g., consistent terminology, acronym expansion). LLMs did *not* contribute to research ideation, dataset or method design, experimental planning, implementation, analysis, or conclusions. All technical content, experiment setup, results, and interpretations were created and verified by the authors, who take full responsibility for the paper's substance and correctness.

## E  REWARD SHAPING AND POLICY INVARIANCE

We treat Stage 3 as the ground-truth MDP, where the terminal reward is

$$r^{(3)} = r_{\text{schema}}\, r_{\text{pass}} \in \{0, 1\}.$$

Hence $r^{(3)} = 1$ if and only if the plan is schema compliant and all commonsense and hard constraints are satisfied, and 0 otherwise. The shaped rewards for Stage 1 and Stage 2 are

$$r^{(1)} = r_{\text{schema}}\big(r_{\text{cs}}^{\text{micro}} + r_{\text{hard}}^{\text{micro}} + r_{\text{cs}}^{\text{macro}} + r_{\text{hard}}^{\text{macro}} + r_{\text{pass}}\big),$$

$$r^{(2)} = r_{\text{schema}}\big(r_{\text{cs}}^{\text{macro}} + r_{\text{hard}}^{\text{macro}} + r_{\text{pass}}\big).$$

We assume throughout that for every initial state (task instance) there exists at least one plan that is schema compliant and satisfies all commonsense and hard constraints.

**Lemma 1.** *Assume that for every instance there exists at least one plan that is schema compliant and satisfies all commonsense and hard constraints. Let $r^{(3)} = r_{schema}\, r_{pass}$ be the ground-truth reward, and let $r^{(1)}, r^{(2)}$ denote the Stage 1 and Stage 2 rewards. Then any optimal policy with respect to $r^{(1)}$ or $r^{(2)}$ attains the ground-truth optimal value under $r^{(3)}$.*

*Proof.* A plan is *valid* if it is schema compliant and all constraints pass. Such a plan yields

$$r^{(3)} = 1, \qquad r^{(1)} = 5, \qquad r^{(2)} = 3,$$

while any invalid plan yields

$$r^{(3)} = 0, \qquad r^{(1)} < 5, \qquad r^{(2)} < 3.$$

Since a valid plan exists, there is a policy that outputs valid plans with probability 1, and therefore $V_*^{(3)} = 1$. Let $\pi_*^{(k)}$ be an optimal policy for reward $r^{(k)}$, with corresponding optimal value of $V_*^{(k)}$. If $\pi_*^{(k)}$ produced an invalid plan with positive probability, its value $V_*^{(k)}$ would be a strict convex combination of the valid and invalid rewards, and thus strictly less than the maximal valid reward (5 for $k = 1$, 3 for $k = 2$). This contradicts optimality. Therefore $\pi_*^{(k)}$ must output valid plans with probability 1. Since $r^{(3)}$ equals 1 exactly for valid plans and 0 otherwise, such a policy satisfies

$$V_{\pi_*^{(k)}}^{(3)} = V_*^{(3)} = 1.$$

Thus any Stage 1 or Stage 2 optimal policy achieves the ground-truth optimal value under $r^{(3)}$.  □

## F  DETAILED CONSTRAINTS

The constraints used in our environment are identical to those in TRAVELPLANNER (Xie et al., 2024). We group them into commonsense and hard constraints as summarized below.

**Commonsense Constraint**

| | |
|---|---|
| Within Sandbox | All information in the plan must be within the closed sandbox; otherwise, it will be considered a hallucination. |
| Complete Information | No key information should be left out of the plan, such as the lack of accommodation during travel. |
| Within Current City | All scheduled activities for the day must be located within that day's city(s). |
| Reasonable City Route | Changes in cities during the trip must be reasonable. |
| Diverse Restaurants | Restaurant choices should not be repeated throughout the trip. |
| Diverse Attractions | Attraction choices should not be repeated throughout the trip. |
| Non-conf. Transportation | Transportation choices within the trip must be reasonable. For example, having both "self-driving" and "flight" would be considered a conflict. |
| Minimum Nights Stay | The number of consecutive days spent in a specific accommodation during the trip must meet the corresponding required minimum number of nights' stay. |

**Hard Constraint**

| | |
|---|---|
| Budget | The total budget of the trip. |
| Room Rule | Room rules include "No parties", "No smoking", "No children under 10", "No pets", and "No visitors". |
| Room Type | Room types include "Entire Room", "Private Room", "Shared Room", and "No Shared Room". |
| Cuisine | Cuisines include "Chinese", "American", "Italian", "Mexican", "Indian", "Mediterranean", and "French". |
| Transportation | Transportation options include "No flight" and "No self-driving". |

# G  BETTER CONVERGENCE SPEED FOR DENSE REWARD

Let's focus on comparing stage 1 reward (dense) with stage 3 reward (sparse).

We compare the optimization dynamics induced by the dense stage–1 reward $\lambda^{(1)} = (1, 1, 1, 1, 1)$ and the sparse stage–3 reward $\lambda^{(3)} = (0, 0, 0, 0, 1)$ from equation 1. For clarity, we write these two terminal rewards as

$$r^{(1)}(u, \tau) := r_{\text{schema}}(u, \tau)\Big(r_{\text{cs}}^{\text{micro}}(u, \tau) + r_{\text{hard}}^{\text{micro}}(u, \tau) + r_{\text{cs}}^{\text{macro}}(u, \tau) + r_{\text{hard}}^{\text{macro}}(u, \tau) + r_{\text{pass}}(u, \tau)\Big), \tag{3}$$

$$r^{(3)}(u, \tau) := r_{\text{schema}}(u, \tau)\, r_{\text{pass}}(u, \tau), \tag{4}$$

for a prompt $u$ and trajectory $\tau$. By construction, $r^{(3)}$ coincides with the original sparse reward, while $r^{(1)}$ is a proper shaping of $r^{(3)}$ (i.e., both induce the same optimal policy).

Let $S$ denote the finite set of training prompts, and let $\pi_{\theta(0)}$ be the initial (tabular or autoregressive) policy. For a generic terminal reward $r$, define the per-prompt reward variance at initialization as

$$V_r(u) := \text{Var}_{\tau \sim \pi_{\theta(0)}(\cdot | u)}\big[\, r(u, \tau)\,\big], \qquad \bar{V}_r := \mathbb{E}_{u \sim S}\big[V_r(u)\big]. \tag{5}$$

**Lemma 2** (Dense reward has larger initial variance). *Assume that, for every prompt $u \in S$,*

*(a) the auxiliary terms are non-negative and non-degenerate under $\pi_{\theta(0)}$:*

$$Z(u, \tau) := r_{schema}(u, \tau)\Big(r_{cs}^{micro} + r_{hard}^{micro} + r_{cs}^{macro} + r_{hard}^{macro}\Big)(u, \tau) \geq 0,$$

*(b) the auxiliary signal is positively correlated with final-pass success:* $\text{Cov}_\tau[\,Z(u, \tau), r_{pass}(u, \tau)\,] \geq 0.$

*Then for every $u \in S$,*

$$V_{r^{(1)}}(u) \;\geq\; V_{r^{(3)}}(u),$$

*and consequently $\bar{V}_{r^{(1)}} \geq \bar{V}_{r^{(3)}}$.*

*Proof.* By definition,

$$r^{(1)}(u, \tau) = r^{(3)}(u, \tau) + Z(u, \tau).$$

For a fixed $u$, write $X := r^{(3)}(u, \tau)$ and $Z := Z(u, \tau)$. Assumption (a) implies $\mathrm{Var}[Z] > 0$, and assumption (b) implies $\mathrm{Cov}[X, Z] \geq 0$. Using $\mathrm{Var}[X + Z] = \mathrm{Var}[X] + \mathrm{Var}[Z] + 2\,\mathrm{Cov}[X, Z]$, we obtain

$$V_{r^{(1)}}(u) = \mathrm{Var}[X + Z] = \mathrm{Var}[X] + \mathrm{Var}[Z] + 2\,\mathrm{Cov}[X, Z] \;\geq\; \mathrm{Var}[X] = V_{r^{(3)}}(u).$$

Averaging over $u \sim S$ gives $\bar{V}_{r^{(1)}} \geq \bar{V}_{r^{(3)}}$. $\qquad\square$

We now connect reward variance to convergence speed using Theorem 1 in (Razin et al., 2025), which shows that the time required for the expected reward to increase is lower bounded by a term inversely proportional to the initial reward variance:

**Theorem 1** (Reward variance controls optimization time, informal). *Under the regularity assumptions of (Razin et al., 2025), there exists a constant $C_\gamma > 0$ (depending on $\gamma$ and problem parameters, but* not *on the choice of $r$) such that for any terminal reward $r$,*

$$t_\gamma(r) \;\geq\; \frac{C_\gamma}{\bar{V}_r}. \tag{6}$$

Applying Theorem 1 to the dense and sparse rewards and combining with Lemma 2 yields:

**Corollary 1** (Dense shaping converges faster than sparse reward, at the level of the lower bound). *Suppose the assumptions of Lemma 2 and Theorem 1 hold. Then*

$$t_\gamma\big(r^{(1)}\big) \;\leq\; \frac{\bar{V}_{r^{(3)}}}{\bar{V}_{r^{(1)}}}\, t_\gamma\big(r^{(3)}\big) \;\leq\; t_\gamma\big(r^{(3)}\big), \tag{7}$$

*whenever $\bar{V}_{r^{(1)}} \geq \bar{V}_{r^{(3)}}$. In particular, the dense stage–1 reward admits a strictly smaller (theoretical) lower bound on optimization time than the sparse stage–3 reward.*

Intuitively, the shaped reward $r^{(1)}$ spreads probability mass across many intermediate scores (partial schema satisfaction, partial commonsense and hard-constraint satisfaction), thereby inducing higher reward variance under the initial policy. By Theorem 1, this larger variance directly reduces the worst-case time required for policy gradient (and its stochastic approximation via GRPO in Section 2.3) to achieve a fixed improvement $\gamma$ in expected final-pass performance. This formalizes the empirical observation that dense, properly shaped rewards lead to faster convergence than the original sparse reward $r^{(3)}$.

## H   Asymptotic performance equivalence between sparse and dense rewards

We now show that, although the dense stage–1 reward $r^{(1)}$ accelerates optimization compared to the sparse stage–3 reward $r^{(3)}$, both induce the *same* asymptotic policy under GRPO.
For $k \in \{1, 3\}$, define the population objective

$$J^{(k)}(\theta) := \mathbb{E}_{u \sim S} \mathbb{E}_{\tau \sim \pi_\theta(\cdot | u)} \big[ r^{(k)}(u, \tau) \big], \tag{8}$$

where $r^{(1)}$ and $r^{(3)}$ are the dense and sparse terminal rewards defined in Section 2.2, and $S$ is the finite set of training prompts.
We assume:

(A1) (*Proper shaping and local shared optimum*) There exists a parameter vector $\theta^\star$ and a neighborhood $U \subset \Theta$ of $\theta^\star$ such that $\theta^\star$ is the *unique* maximizer of both objectives restricted to $U$:

$$\theta^\star = \arg\max_{\theta \in U} J^{(3)}(\theta) = \arg\max_{\theta \in U} J^{(1)}(\theta).$$

This is a local version of proper shaping: near $\theta^\star$ the dense reward $r^{(1)}$ and sparse reward $r^{(3)}$ induce the same optimal policy, but we do not require global uniqueness outside $U$.

(A2) (*Local agreement of GRPO directions via normalized advantages*) There exists a (possibly smaller) neighborhood $U' \subseteq U$ such that for any $\theta \in U'$ and any prompt $u \in S$, the dense and sparse returns differ by a positive affine transform on almost all trajectories sampled from $\pi_\theta(\cdot \mid u)$:

$$r^{(1)}(u, \tau) = a_\theta(u)\, r^{(3)}(u, \tau) + b_\theta(u) \quad \text{with } a_\theta(u) > 0 \quad \text{for } \pi_\theta\text{-almost every } \tau.$$

Since GRPO uses group-normalized returns, this implies that the group-normalized advantages coincide:

$$\hat{A}_i^{(1)} = \hat{A}_i^{(3)} \quad \text{for all } i \text{ and all groups in } U' \text{ (with probability 1.)},$$

and hence the *expected* GRPO gradient fields agree locally:

$$g^{(1)}(\theta) := \mathbb{E}[\nabla_\theta J_{\text{GRPO}}^{(1)}(\theta)] = \mathbb{E}[\nabla_\theta J_{\text{GRPO}}^{(3)}(\theta)] =: g^{(3)}(\theta), \quad \forall \theta \in U'.$$

This is due to the GRPO advantage normalization. Recall that for reward $r^{(k)}$ and a group of trajectories $\{r_i^{(k)}\}_{i=1}^G$ for a fixed prompt $u$, the group-normalized advantage is

$$\hat{A}_i^{(k)} = \frac{r_i^{(k)} - \mu_r^{(k)}}{\sigma_r^{(k)}}, \qquad \mu_r^{(k)} = \frac{1}{G}\sum_{j=1}^G r_j^{(k)}, \quad (\sigma_r^{(k)})^2 = \frac{1}{G}\sum_{j=1}^G (r_j^{(k)} - \mu_r^{(k)})^2. \quad (9)$$

(A3) (*Local convergence of GRPO*) Starting from a common initialization $\theta_0$, the GRPO iterates $\{\theta_N^{(k)}\}_{N\geq 0}$ for $k \in \{1,3\}$ enter $U'$ with probability tending to 1 and remain there thereafter. Moreover, within $U'$ the stochastic GRPO dynamics form a standard stochastic approximation to the ODE $\dot{\theta} = g^{(k)}(\theta)$, and $\theta^\star$ is a locally asymptotically stable equilibrium of this ODE. In particular,

$$\theta_N^{(k)} \xrightarrow{p} \theta^\star \quad \text{as } N \to \infty \text{ for } k \in \{1,3\},$$

but we do *not* require global consistency of GRPO outside $U'$.

Under these assumptions we obtain:

**Theorem 2** (Asymptotic performance equivalence). *Let $\hat{\pi}_N^{(1)} := \pi_{\theta_N^{(1)}}$ and $\hat{\pi}_N^{(3)} := \pi_{\theta_N^{(3)}}$ be the policies obtained after $N$ GRPO updates using the dense and sparse rewards, respectively. Then for every $\varepsilon > 0$,*

$$\mathbb{P}\Big(\big\|\hat{\pi}_N^{(1)} - \hat{\pi}_N^{(3)}\big\| > \varepsilon\Big) \longrightarrow 0 \quad \text{as } N \to \infty, \quad (10)$$

*or equivalently,*

$$\mathbb{P}\Big(\big\|\hat{\pi}_N^{(1)} - \hat{\pi}_N^{(3)}\big\| < \varepsilon\Big) \longrightarrow 1. \quad (11)$$

*Proof sketch.* By (A1) and (A3), for each $k \in \{1,3\}$ the GRPO iterates enter $U'$ and converge in probability to the locally stable maximizer $\theta^\star$ of $J^{(k)}$. Assumption (A2) implies that within $U'$ the expected GRPO update directions coincide: $g^{(1)}(\theta) = g^{(3)}(\theta)$ for all $\theta \in U'$, since dense and sparse rewards differ only by a positive affine transform and group-normalized advantages are invariant under such transforms.

Thus, once both processes have entered $U'$, they follow (up to stochastic noise) the same limiting ODE $\dot{\theta} = g(\theta)$ with the same locally attractive fixed point $\theta^\star$. By standard stochastic approximation arguments, this yields $\theta_N^{(k)} \xrightarrow{p} \theta^\star$ and hence $\hat{\pi}_N^{(k)} = \pi_{\theta_N^{(k)}} \xrightarrow{p} \pi^\star$. Finally, for any $\varepsilon > 0$, applying the triangle inequality and a union bound as in the original argument gives

$$\mathbb{P}\Big(\big\|\hat{\pi}_N^{(1)} - \hat{\pi}_N^{(3)}\big\| > \varepsilon\Big) \xrightarrow[N\to\infty]{} 0,$$

which proves equation 10. $\qquad\qquad\square$

## I  RESPONSE LENGTH CLIPPING ANALYSIS

Figure A.13 shows the evolution of agent response length across training steps for one run. The model often reaches the model output context budget of 30,500 tokens (within a 32K full-context window, where roughly 2K is reserved for the initial prompt), causing recurrent clipping. Once the context is truncated, the agent loses tool outputs and partial trajectories, which forces it to rely on shorter surviving segments of the conversation history. As training progresses the agent adapts by constructing shorter plans to manage the context which leads to sub-par plans.

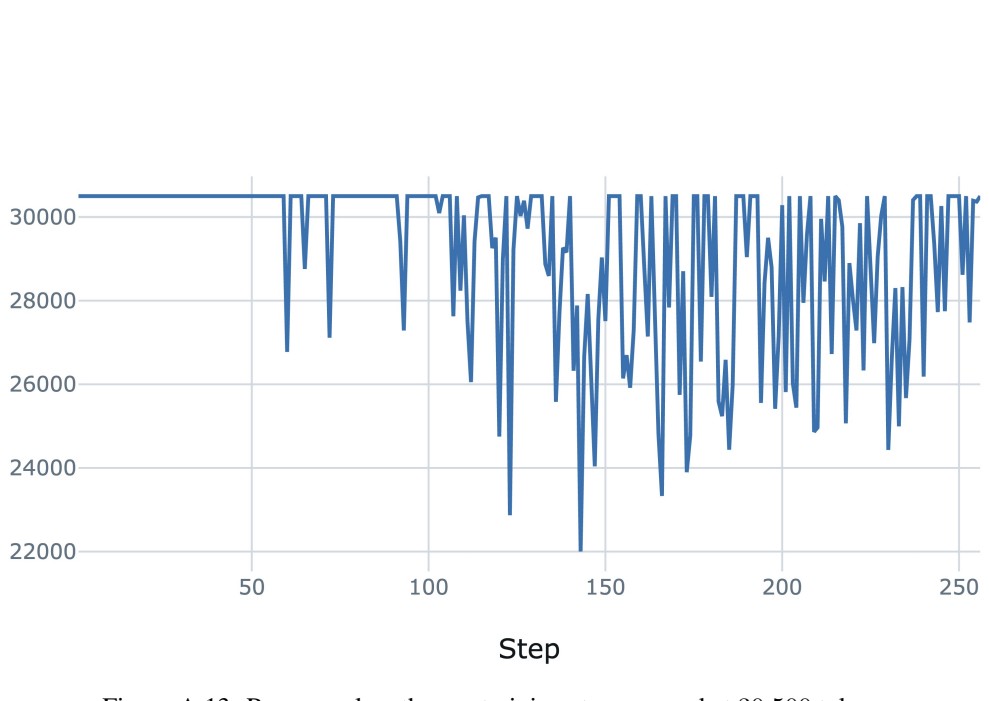

Figure A.13: Response length over training steps capped at 30,500 tokens

