# OpenReview forum: "Planner-R1: Reward Shaping Enables Efficient Agentic RL with Smaller LLMs"
_ICLR.cc/2026/Conference — Submitted to ICLR 2026_

### Official Review · Reviewer_auEt · 2025-10-18

**Soundness:** 3
**Presentation:** 2
**Contribution:** 1
**Rating:** 2
**Confidence:** 4

**Summary:**

The paper targets an important and timely problem — how to use reward shaping to improve the reinforcement learning (RL) process of large language models (LLMs) on long-horizon, tool-based reasoning tasks such as TravelPlanner. The core objective is to **enable smaller LLMs (e.g., 8B models) to achieve competitive performance through efficient post-training**. This aligns well with current practical needs for LLM deployment, where lightweight and efficient models are increasingly desirable.

The authors propose a lightweight combination of multi-stage reward shaping and curriculum learning to make small-scale LLMs learn effectively without depending on large compute budgets. This is a highly meaningful and practically relevant goal.

However, in my view, the current version still feels incomplete as a scientific contribution. While it demonstrates strong performance, it lacks theoretical depth and sufficient justification of generality, making it read more like an engineering case study rather than a full research paper. Of course, a well-executed engineering case study can still be valuable, but in this paper, the experiments are limited to a single domain (TravelPlanner), which **makes it difficult to evaluate whether the proposed method is truly general or simply tailored for this specific environment**.

**Strengths:**

**Strong empirical results**: The performance on the TravelPlanner benchmark is impressive relative to existing methods. The experimental setup is robust, with well-documented configurations, comprehensive ablation studies, and both in-domain and out-of-domain evaluations.

**Practical and relevant direction** : The goal of combining small models with efficient post-training is aligned with current industry deployment constraints (where vast compute is infeasible). The motivation is clear, and the paper addresses an important angle of efficient “agentic” fine-tuning.

**Weaknesses:**

1. **Not clearly distinguishing from existing work** : Reward shaping is a long-established technique in RL, and it’s unclear what fundamentally differentiates the proposed method from prior works. The authors should explicitly explain how their reward scheme differs from previous methods like ORSO: Accelerating Reward Design via Online Reward Selection and Policy Optimization, which also adapts reward functions across training stages. (Of course, you target on a different task, as they just pay attention on the traditional RL setting, you focus on LLM post-training). As currently written, this work largely reuses existing ideas (multi-stage shaping, curriculum reward schedules) without introducing new algorithmic or theoretical insights.

2. **Missing theoretical justification**: The paper introduces a multi-stage reward design that spans from fine-grained micro-level reward components to sparse terminal rewards. However, there is no theoretical guarantee that the shaped reward preserves optimal-policy consistency. Alternatively, the authors could have shown that the proposed reward achieves provably better convergence speed or asymptotic performance than sparse rewards. Without such evidence, this work resembles an empirical best-practice report, not a complete academic study. (An empirical best-practice report can also be valuable if the method demonstrates general applicability across multiple tasks. However, since the proposed approach is evaluated only on the TravelPlanner benchmark, its generalizability remains unclear -- which I discuss in weakness 3)

3. **Generalization and scope are insufficient**: While demonstrating strong results on TravelPlanner, the method is validated on only one domain. This makes it hard to claim generality. From an engineering standpoint, designing a handcrafted reward function for a specific domain can always yield improvement; what makes it academically meaningful is showing that the reward design principle generalizes. The paper does not currently establish that generality.

**Questions:**

I have mentioned the major weakness and related questions in the weakness. Here is some additional small but also important questions.

1. **Reward specification lacks clarity**: In Section 2.2 (“Multi-stage reward”), the authors define several reward components (e.g., \(r^{micro}_{hard}\): fraction of satisfied hard constraints), but they never specify *which constraints* these rewards cover, nor how the constraint satisfaction is computed.

2. **Confusion in stage definitions**: While Section 2.2 defines six distinct reward components, the training procedure includes only three stages (dense, category-level, and sparse). The rationale for maintaining six separate sub-rewards is unclear—for instance, if $(r_{cs}^{micro}\)$ and $(r_{hard}^{micro}\)$ are consistently enabled or disabled together, defining them separately seems redundant.

3. **Confusion in training**: Moreover, Section 2.3 (Optimization) never explains how or when the model transitions between these stages during training. The details of this curriculum scheduling (trigger condition, number of steps, etc.) should be explicitly clarified.

---

> ### Author Response · Authors · 2025-11-21
> **Part-1**
>
> We thank the reviewer for the detailed and constructive feedback. Below we quote each concern and then respond.
>
> > **W1: Not clearly distinguishing from existing work:**
>
> Thank you for raising this point and for highlighting ORSO, which helps clarify the intended scope of our contribution. Our work does not aim to introduce a new reward-shaping algorithm; instead, we intentionally adopt a simple, fixed shaping strategy to examine its practical impact in LLM post-training. ORSO provides an automated approach to reward selection, and we now explicitly acknowledge it as an important future direction in our limitations section:
>
> “**Our method also assumes access to dense, process-level rewards, and its effectiveness may drop in settings with only sparse signals. Recent directions help soften this requirement by automating reward selection methods (Zhang et al., 2025a), using prescribed LLM-as-judge approaches (Lu et al., 2025), or using structured reward design for tool selection (Qian et al., 2025).**”
>
> The core contribution of our paper is empirical guidance, showing that dense, structured rewards enable small models to match or surpass much larger models while using far less compute. This addresses a central challenge in the community, where effective post-training is often assumed to require very large models and substantial GPU resources. In this context, demonstrating that careful reward design makes small-model post-training competitive is a meaningful and actionable insight. Automated methods like ORSO complement this direction but fall outside the scope of the present study.
>
> > **W2: Missing theoretical justification** (same as R3-W1-Theoretical)
>
>  We appreciate the request for stronger theoretical grounding. In the revision, the appendix now includes 3 theoretical contributions: 1) The optimal policies for Stages 3 and 2 are optimal for Stage 1 reward (Appendix E), 2) faster convergence under dense rewards due to reduced gradient variance (Appendix G), 3) Asymptotic equivalence showing shaped dense reward preserves resulting policy consistency under GRPO training (Appendix H).
>
> > **W3: Generalization and scope are insufficient** (Same as R3-W1-Benchmark difficulty)
>
> We appreciate the reviewer’s concern. The goal of this paper is not to introduce a new reward-shaping algorithm, but to provide the insight that **small models can be strong and far more compute-efficient agentic RL learners when dense, well-structured process rewards are available**. With this framing, we agree that evaluating additional domains would further strengthen generality. However, our results are consistent with a broader empirical trend: **7–8B models approach the performance of 30–32B models** in simpler domains. Examples include:
>
> | Dataset | Small Model | Large Model | Small Perf. | Large Perf. | Paper |
> |--------|-------------|-------------|-------------|-------------|--------|
> | HumanEval (pass@1) | Qwen2.5-Coder-7B-Instruct | Qwen2.5-Coder-32B-Instruct | 88.4% | 92.7% | arXiv:2409.12186 |
> | GSM8K (CoT), 8-shot | Phi-3 Small (7B) | Phi-3 Medium (14B) | 89.6% | 91% | arXiv:2404.14219 |
>
> Beyond this general trend, **TRAVELPLANNER is already a highly challenging agentic environment**, not a toy domain. Even GPT-5 (high reasoning effort) achieves only about 20% final-pass accuracy. The agent must interpret complex travel intents, autonomously call APIs, handle missing information, and satisfy dozens of constraints across trajectories that often exceed 30.5k tokens. These properties reflect real-world long-horizon planning much more closely than synthetic benchmarks.
>
> To ensure robustness and reproducibility, we ran **five independent seeds per configuration**, all using identical scripts, random seeds, curriculum schedules, and reward setups. All runs were executed on **two H200 nodes (16 GPUs)** with full logging and fixed pipelines to eliminate infrastructural variance. Running five seeds for 500 steps for both 8B and 32B models cost approximately **$2K** and **$7K** respectively, bringing the minimum cost of the study to **$67K** across all configurations. Additional exploratory sweeps roughly doubled this footprint. Given the importance of long-horizon planning tasks for agentic RL, we view this investment as both meaningful and necessary.
>
> The key contribution of this work is empirical guidance showing that dense, structured rewards enable small models to match or surpass much larger models while using far less compute. This challenges the common assumption that effective post-training for long-horizon agent tasks requires large models and large GPU budgets, and provides a practical path for resource-constrained practitioners.

---

> ### Author Response · Authors · 2025-11-21
> **Part-2**
>
> > **Q1: Reward specification lacks clarity**
>
> We agree that clarity around the exact constraint sets and how each reward component is computed is important for reproducibility. As noted in paper, *“Formal definitions and the complete list of constraints are provided in the work of Xie et al. (2024).”* TravelPlanner comes with an established, executable evaluation suite, and our reward components (micro-level and macro-level) are computed directly using this official constraint checker.
> For reader's convenience, we have now also included the *full, explicit description of all commonsense and hard constraints* in **Appendix F**.
>
> > **Q2: Confusion in stage definitions**
>
> Our decision to define six reward components but use only three training stages follows the structure of the original TravelPlanner framework. We kept the micro/macro and hard/commonsense distinctions because they represent different types of constraint signals, even if some components are enabled together during training. That said, other staging configurations, such as activating hard and commonsense constraints independently, are also reasonable. We chose the three-stage setup to keep the experimental space tractable given the high cost of multi-seed GRPO runs. Automated approaches like ORSO as you highlighted could further explore these alternatives more efficiently, which we see as a natural direction for future work.
>
> > **Q3: Confusion in training**
>
> The transition schedule is specified in three different parts of the submission below. Please let us know if you need further details.
>
> - **Section 2.2 (line 207):** *“transitions occur at predefined step counts.”*
> - **Section 3.1 (lines 239–241):** *“preserving the easy/medium/hard ratio. We evaluated three single-stage reward configurations with 500 steps and a curriculum regime for 8B and 32B models with 100/300/100 and 50/350/100 steps, respectively.”*
> - **Table 1 caption (lines 326–327):** *“Curriculum uses three phases: for 8B, 100/300/100 steps; for 32B, 50/350/100 steps across Stages 1–3.”*

---

### Official Review · Reviewer_zmo9 · 2025-10-31

**Soundness:** 2
**Presentation:** 2
**Contribution:** 2
**Rating:** 4
**Confidence:** 3

**Summary:**

This paper investigates agentic reinforcement learning (RL) with large language models (LLMs) on the TravelPlanner benchmark. The proposed approach, Planner-R1, achieves a 56.9% final-pass rate with only 180 training queries, representing a 2.7× improvement over GPT-5’s baseline. A key finding is that smaller models (8B) perform competitively with dense reward shaping, achieving better compute and memory efficiency compared to larger models (32B). While larger models are more robust under sparse rewards, smaller models show greater responsiveness to dense shaping. Shaped rewards enhance learning dynamics without compromising generalization, as evidenced by strong out-of-domain performance. The paper concludes that reward shaping is a decisive factor for scaling agentic RL efficiently.

**Strengths:**

1. The study provides a new empirical finding regarding reward shaping’s role in improving the efficiency of smaller models, which could inspire future work in agentic RL.
2. Despite being empirical in nature, the paper conducts a significant amount of experimentation, including out-of-domain evaluations, to support its conclusions.

**Weaknesses:**

1. The paper primarily focuses on empirical observations without proposing fundamentally new methods or theoretical insights, making it more of an empirical study than a groundbreaking contribution.
2. The findings, while interesting, lack sufficient explanation or justification for why smaller models benefit more from reward shaping, why larger models show higher variance, and why curriculum learning offers no benefit. These gaps leave the conclusions less convincing and potentially unstable.
3. While the paper claims that fine-tuned models generalize well, the results could still reflect subtle overfitting tendencies, especially given the high variance in larger models and the seemingly counterintuitive behavior compared to broader trends in scaling laws for LLMs.
4. Only 8B and 32B are tested in detail while larger size models are not included. Only two sizes may not be enough to get the conclusion.

**Questions:**

1. How stable are the findings across different seeds, training runs, or benchmarks? Given the high variance in larger models, what steps were taken to ensure the observed improvements are not the result of chance?
2. The observation that smaller models (8B) are more efficient and competitive under dense rewards seems to contradict the general trend of larger models excelling in complex tasks. Did you observe any limitations of smaller models in other settings, or do these results suggest a broader pattern?

---

> ### Author Response · Authors · 2025-11-21
> **Part-1**
>
> Thank you for the constructive and detailed feedback, genuinely appreciate the effort in highlighting both the strengths and the areas where deeper analysis and clarification will meaningfully improve the paper.
>
> > **W1: The paper primarily focuses on empirical observations without proposing fundamentally new methods or theoretical insights, making it more of an empirical study than a groundbreaking contribution.**
>
> **Theoretical Justification**: We appreciate the request for stronger theoretical grounding. In the revision, the appendix now includes 3 theoretical contributions: 1) The optimal policies for Stages 3 and 2 are optimal for Stage 1 reward (Appendix E), 2) faster convergence under dense rewards due to reduced gradient variance (Appendix G), 3) Asymptotic equivalence showing shaped dense reward preserves resulting policy consistency under GRPO training (Appendix H).
>
> **Benchmark Difficulty, Experimental Rigor, Generalizability**:
>
> We appreciate the reviewer’s concern. The goal of this paper is not to introduce a new reward-shaping algorithm, but to provide the insight that **small models can be strong and far more compute-efficient agentic RL learners when dense, well-structured process rewards are available**. With this framing, we agree that evaluating additional domains would further strengthen generality. However, our results are consistent with a broader empirical trend: **7–8B models approach the performance of 30–32B models** in simpler domains. Examples include:
>
> | Dataset | Small Model | Large Model | Small Perf. | Large Perf. | Paper |
> |--------|-------------|-------------|-------------|-------------|--------|
> | HumanEval (pass@1) | Qwen2.5-Coder-7B-Instruct | Qwen2.5-Coder-32B-Instruct | 88.4% | 92.7% | arXiv:2409.12186 |
> | GSM8K (CoT), 8-shot | Phi-3 Small (7B) | Phi-3 Medium (14B) | 89.6% | 91% | arXiv:2404.14219 |
>
> Beyond this general trend, **TRAVELPLANNER is already a highly challenging agentic environment**, not a toy domain. Even GPT-5 (high reasoning effort) achieves only about 20% final-pass accuracy. The agent must interpret complex travel intents, autonomously call APIs, handle missing information, and satisfy dozens of constraints across trajectories that often exceed 30.5k tokens. These properties reflect real-world long-horizon planning much more closely than synthetic benchmarks.
>
> To ensure robustness and reproducibility, we ran **five independent seeds per configuration**, all using identical scripts, random seeds, curriculum schedules, and reward setups. All runs were executed on **two H200 nodes (16 GPUs)** with full logging and fixed pipelines to eliminate infrastructural variance. Running five seeds for 500 steps for both 8B and 32B models cost approximately **$2K** and **$7K** respectively, bringing the minimum cost of the study to **$67K** across all configurations. Additional exploratory sweeps roughly doubled this footprint. Given the importance of long-horizon planning tasks for agentic RL, we view this investment as both meaningful and necessary.
>
> **Core Contribution and Significance**: The key contribution of this work is empirical guidance showing that dense, structured rewards enable small models to match or surpass much larger models while using far less compute. This challenges the common assumption that effective post-training for long-horizon agent tasks requires large models and large GPU budgets, and provides a practical path for resource-constrained practitioners.

---

> ### Author Response · Authors · 2025-11-21
> **Part-2**
>
> > **W2: The findings, while interesting, lack sufficient explanation or justification for why smaller models benefit more from reward shaping, why larger models show higher variance, and why curriculum learning offers no benefit. These gaps leave the conclusions less convincing and potentially unstable.**
>
> **Why smaller models benefit more from reward shaping. Why curriculum learning offers no benefit**:  (Same as R2-W2)
>
> The behavior of the 8B model under sparse Stages 2 and 3 rewards is expected. With only terminal success signals, the 8B model rarely produces any correct early trajectories, which yields near-zero advantages and prevents learning within a fixed compute budget. Dense process rewards resolve this by providing graded feedback across micro (easy), macro (medium), and full pass rate (hard) components, allowing the 8B model to improve steadily without requiring full task success. The 32B model, with a much stronger initial policy, can generate partial solutions even under sparse rewards, making it naturally less dependent on reward density.
>
> This same structure explains why curriculum learning did not provide further gains. The Stage 1 dense reward already behaves as a soft curriculum. Its multi-level components lead the model to shift attention from easy to hard objectives as the former saturate. This implicit progression is consistent with prior findings such as ToolRL, which shows that gradual reward changes outperform hard switches. Under our static and heuristic schedule, explicit transitions offered no additional benefit. For the 8B model, the switch occurred too early and reduced non-zero learning signal. For the 32B model, all stages produced similar gradients, so switching had some improvement but it was not significant.
>
> We do not claim that curriculum learning is ineffective in general. Our results show that in this environment, dense rewards alone were sufficient and explicit stages did not improve over the implicit curriculum already provided by Stage 1. We clarify these in the paper and now make explicit in the paper that our approach relies on access to dense, process-level rewards. This limits applicability in settings with only sparse signals, and we have updated the Limitations section to state this clearly. We also highlight emerging methods such as ORSO and LLM-as-judge reward construction as promising ways to obtain dense supervision in new domains.
> We made the following changes to the paper:
>
> - **Evalutions**: “...The 8B model was highly sensitive to reward sparsity: Stage 2 and Stage 3 led to 3/5 and 5/5 collapses, consistent with prior evidence that smaller models depend strongly on shaped feedback (Ng et al., 1999; dos Santos et al., 2024; Qian et al., 2025). These collapses are expected. Under sparse Stage 2 and Stage 3 rewards, the 8B model rarely produces correct early trajectories, yielding near-zero advantages and stalling learning within the available compute budget, which also explains the wide confidence intervals for Stage 2 and the zero performance at Stage 3. Dense process rewards supply graded feedback across micro, macro, and final-pass components, allowing steady improvement and effectively acting as a soft curriculum. This implicit progression aligns with ToolRL (Qian et al., 2025), which shows that gradual reward changes outperform hard switches, and explains why explicit curriculum transitions added no measurable benefit in our setup."
>
> - **Limitations**: “We focused on TRAVELPLANNER, a constrained benchmark, and smaller models may not remain competitive on more complex or open-ended tasks. Although 8B models are more FLOP-efficient, larger models still achieve higher peak accuracy and robustness, which can be essential when reliable intermediate feedback is unavailable. Our method also assumes access to dense, process-level rewards, and its effectiveness may drop in settings with only sparse signals. Recent directions help soften this requirement by automating reward selection methods (Zhang et al., 2025a), using prescribed LLM-as-judge approaches (Lu et al., 2025), or employing structured reward design for tool selection (Qian et al., 2025). Finally, we used a fixed three-stage curriculum, which did not yield clear advantages, and exploring adaptive or automated scheduling strategies remains an important direction for future work. Overall, our results showed that with reliable dense feedback, reward shaping enables smaller models to become strong and compute-efficient alternatives.”

---

> ### Author Response · Authors · 2025-11-21
> **Part-3**
>
> **Why larger models show higher variance**:  (Same as R2-Q4)
>
> TRAVELPLANNER is a challenging environment where essential information is often missing, requiring the agent to adapt, recover, and explore alternative solution paths. Our hypothesis is that smaller models explore less and behave more deterministically, as their limited capacity concentrates probability mass on a narrow set of actions, even when those actions are suboptimal. Larger models, by contrast, can consider a broader set of strategies under uncertainty (for example, switching transportation modes or hotel choices when details are missing). We therefore attribute the higher variance of the 32B model to its greater exploratory behavior during RL, amplified by the inherently uncertain nature of the task.
>
> We reflected that in the evaluation section:
> “Although the 32B model attained a slightly higher peak accuracy (56.9% vs. 56.4%), this difference was not statistically significant and was accompanied by higher variance. **We hypothesize that this variance stems from differences in exploration dynamics. TRAVELPLANNER is a challenging environment where essential information is often missing, requiring recovery and alternative solution paths. Smaller models tend to explore less and behave more deterministically, concentrating probability mass on a narrow set of actions. Larger models, by contrast, can consider a broader range of strategies under uncertainty (for example, switching transportation modes or hotel choices), leading to higher variability across rollouts. This exploratory flexibility likely contributes to the increased variance observed in the 32B runs**.”
>
> > **W3: While the paper claims that fine-tuned models generalize well, the results could still reflect subtle overfitting tendencies, especially given the high variance in larger models and the seemingly counterintuitive behavior compared to broader trends in scaling laws for LLMs.** (Same as R2-W3)
>
> Appreciate raising this concern. We further updated the paper as follows:
>
> - **Introduction**: “...Planner-R1 models maintained or exceeded baseline performance on out-of-domain tasks including Multi-IF, NaturalPlan, and τ-Bench, **provided that excessive domain-specific fine-tuning is avoided.**”
>
> - **Abstract**: “... fine-tuned models **mostly** maintained or exceeded baseline performance on out-of-domain tasks, including Multi-IF, NaturalPlan, and τ-Bench.”
>
> - **Limitation**: “At 3,000 steps, the 8B model improved five of seven metrics but regressed on two, **indicating the risk of
> excessive fine tuning.**”
>
> > **W4: Only 8B and 32B are tested in detail while larger size models are not included. Only two sizes may not be enough to get the conclusion.**
>
> We agree that results across additional model sizes would be valuable. We did try the 4B model, but even with our dense reward it could not achieve any non-zero returns. As the two configurations evaluated here required substantial computational resources, as detailed in our response to W1, we did not explore bigger models.
>
> > **Q1: How stable are the findings across different seeds, training runs, or benchmarks? Given the high variance in larger models, what steps were taken to ensure the observed improvements are not the result of chance?**
>
> To ensure stability, we trained every model configuration across 5 independent runs and likewise evaluated each configuration on the out-of-domain benchmark 5 times. Response quality and reproducibility were priorities in this work, and we followed a consistent training pipeline, and identical infrastructure across all runs. As noted by Reviewer SV5k, “The experimental quality here is strong. The authors report means and 95% CIs over 5 runs, which is respectable.” These repeated trials required substantial computing and reflected deliberate effort to ensure that the reported improvements are robust rather than artifacts of chance.

---

> ### Author Response · Authors · 2025-11-21
> **Part-4**
>
> > **Q2: The observation that smaller models (8B) are more efficient and competitive under dense rewards seems to contradict the general trend of larger models excelling in complex tasks. Did you observe any limitations of smaller models in other settings, or do these results suggest a broader pattern?**
>
> This is an excellent question, and we have added this discussion to the limitations section. Our claim is not that large models are unnecessary, but that an 8B model can approach the performance of a 32B model when dense, well-structured process rewards are available. The 32B model remains more capable overall, and in settings where process rewards are difficult to specify or evaluate, larger models are still the preferred choice. Our results show that when intermediate rewards are clear and reliably computable, smaller models become a viable and far more compute-efficient alternative. This does not contradict the broader trend of larger models excelling on complex tasks; rather, it identifies a regime where reward design enables smaller models to close much of the gap while using substantially fewer FLOPs. At the same time, our findings also highlight a key limitation: smaller models fail under sparse rewards in agentic RL, whereas larger models remain robust.

---

> ### Comment · Reviewer_zmo9 · 2025-11-27
>
> The author's response regarding the theory, model size, and variance has to some extent alleviated my concerns. I have decided to raise the score to 6.

---

### Official Review · Reviewer_SV5k · 2025-11-01

**Soundness:** 3
**Presentation:** 3
**Contribution:** 3
**Rating:** 6
**Confidence:** 3

**Summary:**

Overall, I'm quite positive on this paper. The authors tackle a very timely problem: the trade-off between model size, compute efficiency, and performance in agentic RL. The core claim is fascinating: they show that a smaller 8B model (PLANNER-R1-8B), when trained with dense process-level rewards, can achieve performance (56.4%) on par with a much larger 32B model (56.9%) on the difficult TRAVELPLANNER benchmark, while being 3.5x more compute-efficient. This provides some of the first strong empirical evidence for the "SLM-First" hypothesis in agentic AI.

The paper's strengths are its high significance, strong experimental quality (especially the inclusion of OOD generalization tests), and a very novel analysis of the interaction between model scale and reward density.

However, the paper is held back by two major flaws: (1) It completely fails to cite or discuss several critical, highly relevant prior works that use the exact same algorithms and models, which calls its novelty claims into question. (2) It presents fascinating but negative results (e.g., the failure of curriculum learning and the total collapse of the 8B model on sparse rewards) and then fails to analyze or even discuss them. This second point is a deep intellectual flaw, as it ignores a central contradiction in the paper's own findings.

I am "on the fence" and my final recommendation will depend entirely on how the authors address the major weaknesses and questions below in their rebuttal.

**Strengths:**

1. **Significance:** This is the paper's greatest strength. The field has been debating the role of SLMs vs. LLMs for agentic tasks. This debate was largely hypothetical, kicked off by position papers like the one from NVIDIA (Belcak et al., 2025) which argued that SLMs were the future. This paper, "PLANNER-R1," provides what is, to my knowledge, the first strong empirical evidence that this hypothesis is correct, at least on a complex, constraint-driven task. Figure 3, showing the 8B model achieving 90% of the 32B's peak performance for a fraction of the FLOPs, is a hugely important result for the community. It moves the "SLM-First" idea from a "position" to an "empirically-backed strategy." This is a really big deal.

2. **Quality:** The experimental quality here is strong. The authors report means and 95% CIs over 5 runs, which is respectable. What I especially appreciate is the inclusion of out-of-domain (OOD) generalization tests in Table 2. This is a critical step that many RL-finetuning papers omit. By showing that their models don't suffer catastrophic overfitting and maintain performance on tasks like NATURALPLAN and T-BENCH, they proactively address the most common criticism of this kind of task-specific tuning. The qualitative analysis in Appendix A is also insightful.

3. **Originality:** The originality here is nuanced, but significant. To be clear, the components themselves are not new. The field was already using GRPO and VERL, and the idea of using process-based rewards over outcome-based rewards was already the consensus. The real original contribution, in my opinion, is the paper's analysis of the interaction between model scale (8B vs 32B) and reward density (Stage 1 vs Stage 3).

   The key new piece of knowledge from this paper is the discovery of an asymmetric response to reward shaping. Look at Table 1:

   - The 8B model is extremely sensitive. It gets 39.9% on the dense Stage 1 reward but completely collapses to 0.0% on the sparse Stage 3 reward.
   - The 32B model is robust. It gets 42.3% on Stage 1 and still gets 44.3% on Stage 3.
      This finding—that SLMs have this asymmetric, sensitive response to rewards—is novel and, frankly, very valuable. It tells us how to train SLMs effectively.

4. **Clarity:** The paper is just very well-written. The problem formulation as an MDP in 2.1 is clean. The multi-stage reward function in 2.2 is well-motivated and easy to understand. The figures are clear and directly support the main claims.

**Weaknesses:**

But this is where my major problems start. The paper has two significant flaws that prevent me from recommending acceptance in its current state.

1. **Missing Key Citations & Discussion (Major Flaw):** The paper has a massive blind spot regarding related work, to the point where it seriously undermines the claims of novelty. The authors must address this.

   - First, 'ToolRL' (Qian et al., 2025): This paper is not cited. "ToolRL" is explicitly self-described as the "first comprehensive study on reward design for tool selection... within the RL paradigm". It also uses GRPO and also concludes that "fine-grained reward decomposition leads to more stable and effective learning". This is so similar to this paper's premise that not citing it is a serious omission.
   - Second, 'PilotRL' (Sheng et al., 2025): This paper is also not cited. "PilotRL" uses almost the exact same technical stack: it explicitly trains a Qwen3-8B model using GRPO on the VERL framework for planning tasks.

   **Constructive Suggestion:** The authors must add a paragraph to their Related Work (Section 5) that discusses these papers. They need to clearly differentiate their work. The most obvious differentiator is that "PLANNER-R1" is the first to do a rigorous comparative analysis of reward-density-vs-model-scale (8B vs 32B) on a hard benchmark like TRAVELPLANNER, whereas the other papers focused on either the reward design itself (ToolRL) or the application of the stack (PilotRL). But as written, the paper implies it was the first to do any of this, which is incorrect.

2. **Severe Lack of Analysis on Negative Results (Major Flaw):** My other major, major flaw with this paper is its complete failure to analyze its own negative results. The authors present two fascinating findings in Table 1, and then... just ignore them. This is not good scientific practice.

   - 1. The Failure of Curriculum Learning: The paper states, "Curriculum learning... provided no measurable benefit". This is an understatement. Table 1 shows the 8B-Curriculum model got 27.1%, which is dramatically worse than the simple 8B-Stage1 model's 39.9%. This is highly counter-intuitive; curriculum strategies are supposed to help! Why did it fail so badly? Did the model experience catastrophic forgetting? Was the transition between stages too abrupt? The authors offer zero hypotheses. This is a fascinating negative result, and they just walk past it.
   - 1. The Fragility of the 8B Model: This is the most important one. As I said in 'Strengths', the 8B model's performance completely collapses (0.0% success) when trained on the sparse Stage 3 reward, while the 32B model is fine (44.3%). This reveals a deep and critical limitation to the paper's entire thesis. The authors' "SLM-First" efficiency argument completely depends on having access to expensive, dense, process-level rewards.

   This is a core contradiction that is never discussed. The authors portray SLMs as the cheap option, but their results prove SLMs depend on the most expensive type of reward signal. How do we get these dense rewards for new tasks? That's the billion-dollar question! This limitation is central to the paper's claim and must be discussed in the Limitations section.

3. **Slight Overstatement of Generalization (Minor Flaw):** This is a more minor point, but the abstract and discussion claim the gains came "without sacrificing generalization". However, Table 2 itself shows this isn't quite true. The `Planner-R1-8B (3000 steps)` model shows a statistically significant drop in performance on two OOD tasks: `NATURALPLAN (Trip)` (12.9 down to 10.7) and `NATURALPLAN (Meeting)` (22.7 down to 20.1). This suggests that while moderate finetuning is fine, "excessive fine-tuning" (as the authors call it) does begin to cause overfitting. This is a slight exaggeration and should be toned down to be more precise.

**Questions:**

My questions are aimed at getting the authors to address the weaknesses I've listed.

- **Q1 (On Novelty):** Can you please clarify the exact novel contribution of your work given that 'ToolRL' (Qian et al., 2025) already did a comprehensive study of reward design for tools with GRPO, and 'PilotRL' (Sheng et al., 2025) already applied the Qwen3-8B+GRPO+VERL stack? Am I correct in thinking your core novelty is specifically the 8B-vs-32B comparative analysis on TRAVELPLANNER?
- **Q2 (On Curriculum Learning):** I was fascinated by the failure of your curriculum learning strategy (Table 1), where the 8B-Curriculum (27.1%) performed much worse than the 8B-Stage1 (39.9%). This is a very interesting negative result. Why do you think this happened? Do you have any hypotheses (e.g., catastrophic forgetting)?
- **Q3 (On SLM Reward Dependence):** This is my biggest question. Your results show the 8B model collapses (0.0% performance) on sparse rewards, while the 32B model is robust (44.3%). This implies your impressive efficiency gains for SLMs are entirely dependent on having access to dense, process-level rewards, which are notoriously expensive and difficult to create for new tasks. How do you reconcile this? Doesn't this limit the practical utility of your "SLM-First" approach to only tasks where we can already afford to create dense process rewards?
- **Q4 (On 32B Variance):** You note that the 32B model exhibited "higher variance", and the CIs in Table 1 confirm this (e.g., 44.3 ± 14.1). Why do you think the larger model was less stable during training? That seems counter-intuitive.
- **Q5 (On the Solver Gap):** You correctly position your 56.9% as SOTA for end-to-end agents. But you also note that the hybrid "solver" method (Hao et al., 2025) gets ~94-97% on this same benchmark. Do you believe that this massive ~37% gap can ever be closed by pure RL agent methods (e.g., with more data or better algorithms), or does this gap suggest that for hard-constraint tasks, hybrid neuro-symbolic methods will always be superior?
- **Q6 (On Disabling "Thinking"):** You mention you disabled the Qwen3 `<think>` token due to memory/context constraints. Since this "thinking" is a form of internal monologue (like Chain-of-Thought), is it possible this crippled the 32B model's ability to use the process-level rewards? In other words, perhaps the 32B model needed that "thinking" scratchpad to properly internalize the dense reward signals, and by removing it, you inadvertently made it look less responsive to reward shaping than it truly is?

---

> ### Author Response · Authors · 2025-11-21
> **Part-1**
>
> We thank the reviewer for the careful, detailed, and insightful review. We appreciate the positive assessment of our contributions as well as the constructive feedback on missing citations and analysis gaps. We respond to each concern point-by-point below.
>
> > **W1: Missing citation and insufficient discussion of closely related work.**
>
> We thank the reviewer for pointing out the missing discussion of ToolRL and PilotRL.
> ToolRL [1] focuses on reward design for tool selection under Reinforcement Fine Tuning (RFT). Its domains (ToolACE, xLAM) and tool outputs are fully synthetic: teacher models generate the “ground truth’’ tool calls, and LLMs simulate tool responses. As a result, ToolRL does not interact with a real environment and does not implement an agentic RL process; optimization reduces to token imitation rather than solving a sequential MDP with 7B LLMs.
> PilotRL [2] proposes a planner–executor decomposition (AdaPlan) and uses GRPO + VERL with LLM-as-judge rewards. It relies on DeepSeek-V3 to simulate environment behavior and focuses on modular agent optimization with LLMs up to 8B. However, it does not study how reward density interacts with model scale or how sparsity affects convergence and final-pass performance.
> Our work, as correctly mentioned by you, directly studies this missing dimension: the interaction between reward density and model scale (8B vs 32B) under real tool-calling environments. Using TRAVELPLANNER as a challenging domain with executable APIs, we show that sparse vs dense process rewards have asymmetric effects on small vs large models, revealing the fragility of 8B models and the robustness of 32B models. This scaling-sensitivity analysis is not addressed by ToolRL or PilotRL.
> We extended Section 5 in our revision to include these new information. We also added a recent comprehensive survey (Nov 2025) on Agentic RL for LLMs [3] to our introduction.
>
> [1] Cheng Qian, Emre Can Acikgoz, Qi He, Hongru Wang, Xiusi Chen, Dilek Hakkani-Tür, Gokhan Tur, and Heng Ji, “ToolRL: Reward is All Tool Learning Needs,” in arXiv preprint, Apr. 2025.
>
> [2] Keer Lu, Chong Chen, Bin Cui, Huang Leng, and Wentao Zhang, “PilotRL: Training Language Model Agents via Global Planning-Guided Progressive Reinforcement Learning,” arXiv, Aug. 2025. **We noticed it is Lu et al. 2025, instead of shane et al. 2025. Please let us know if we found the wrong citation.**
>
> [3] G. Zhang, H. Geng, X. Yu, Z. Yin, Z. Zhang, Z. Tan, H. Zhou, Z. Li, X. Xue, Y. Li, Y. Zhou, Y. Chen, C. Zhang, Y. Fan, Z. Wang, S. Huang, Y. Liao, H. Wang, M. Yang, H. Ji, M. Littman, J. Wang and S. Yan, “The Landscape of Agentic Reinforcement Learning for LLMs: A Survey,” arXiv preprint, arXiv:2509.02547, 2025.

---

> ### Author Response · Authors · 2025-11-21
> **Part-2**
>
> > **“W2: The Failure of Curriculum Learning & The Fragility of the 8B Model not discussed.”**
>
> The behavior of the 8B model under sparse Stages 2 and 3 rewards is expected. With only terminal success signals, the 8B model rarely produces any correct early trajectories, which yields near-zero advantages and prevents learning within a fixed compute budget. Dense process rewards resolve this by providing graded feedback across micro (easy), macro (medium), and full pass rate (hard) components, allowing the 8B model to improve steadily without requiring full task success. The 32B model, with a much stronger initial policy, can generate partial solutions even under sparse rewards, making it naturally less dependent on reward density.
>
> This same structure explains why curriculum learning did not provide further gains. The Stage 1 dense reward already behaves as a soft curriculum. Its multi-level components lead the model to shift attention from easy to hard objectives as the former saturate. This implicit progression is consistent with prior findings such as ToolRL, which shows that gradual reward changes outperform hard switches. Under our static and heuristic schedule, explicit transitions offered no additional benefit. For the 8B model, the switch occurred too early and reduced non-zero learning signal. For the 32B model, all stages produced similar gradients, so switching had some improvement but it was not significant.
>
> We do not claim that curriculum learning is ineffective in general. Our results show that in this environment, dense rewards alone were sufficient and explicit stages did not improve over the implicit curriculum already provided by Stage 1. We clarify these in the paper and now make explicit in the paper that our approach relies on access to dense, process-level rewards. This limits applicability in settings with only sparse signals, and we have updated the Limitations section to state this clearly. We also highlight emerging methods such as ORSO and LLM-as-judge reward construction as promising ways to obtain dense supervision in new domains.
> We made the following changes to the paper:
>
> - **Evalutions**: “...The 8B model was highly sensitive to reward sparsity: Stage 2 and Stage 3 led to 3/5 and 5/5 collapses, consistent with prior evidence that smaller models depend strongly on shaped feedback (Ng et al., 1999; dos Santos et al., 2024; Qian et al., 2025). These collapses are expected. Under sparse Stage 2 and Stage 3 rewards, the 8B model rarely produces correct early trajectories, yielding near-zero advantages and stalling learning within the available compute budget, which also explains the wide confidence intervals for Stage 2 and the zero performance at Stage 3. Dense process rewards supply graded feedback across micro, macro, and final-pass components, allowing steady improvement and effectively acting as a soft curriculum. This implicit progression aligns with ToolRL (Qian et al., 2025), which shows that gradual reward changes outperform hard switches, and explains why explicit curriculum transitions added no measurable benefit in our setup."
>
> - **Limitations**: “We focused on TRAVELPLANNER, a constrained benchmark, and smaller models may not remain competitive on more complex or open-ended tasks. Although 8B models are more FLOP-efficient, larger models still achieve higher peak accuracy and robustness, which can be essential when reliable intermediate feedback is unavailable. Our method also assumes access to dense, process-level rewards, and its effectiveness may drop in settings with only sparse signals. Recent directions help soften this requirement by automating reward selection methods (Zhang et al., 2025a), using prescribed LLM-as-judge approaches (Lu et al., 2025), or employing structured reward design for tool selection (Qian et al., 2025). Finally, we used a fixed three-stage curriculum, which did not yield clear advantages, and exploring adaptive or automated scheduling strategies remains an important direction for future work. Overall, our results showed that with reliable dense feedback, reward shaping enables smaller models to become strong and compute-efficient alternatives.”
>
> > “W3: Slight Overstatement of Generalization (Minor Flaw)”
>
> Appreciate raising this concern. We further updated the paper as follows:
>
> - **Introduction**: “...Planner-R1 models maintained or exceeded baseline performance on out-of-domain tasks including Multi-IF, NaturalPlan, and τ-Bench, **provided that excessive domain-specific fine-tuning is avoided.**”
>
> - **Abstract**: “... fine-tuned models **mostly** maintained or exceeded baseline performance on out-of-domain tasks, including Multi-IF, NaturalPlan, and τ-Bench.”
>
> - **Limitation**: “At 3,000 steps, the 8B model improved five of seven metrics but regressed on two, **indicating the risk of
> excessive fine tuning.**”

---

> ### Author Response · Authors · 2025-11-21
> **Part-3**
>
> > **Q1 (On Novelty): Can you please clarify the exact novel contribution of your work given that 'ToolRL' (Qian et al., 2025) already did a comprehensive study of reward design for tools with GRPO, and 'PilotRL' (Sheng et al., 2025) already applied the Qwen3-8B+GRPO+VERL stack? Am I correct in thinking your core novelty is specifically the 8B-vs-32B comparative analysis on TRAVELPLANNER?**
>
> Please refer to the answer to W1.
>
> > **Q2 (On Curriculum Learning): I was fascinated by the failure of your curriculum learning strategy (Table 1), where the 8B-Curriculum (27.1%) performed much worse than the 8B-Stage1 (39.9%). This is a very interesting negative result. Why do you think this happened? Do you have any hypotheses (e.g., catastrophic forgetting)?**
>
> Please refer to the answer to W2.
>
> > **Q3 (On SLM Reward Dependence): This is my biggest question. Your results show the 8B model collapses (0.0% performance) on sparse rewards, while the 32B model is robust (44.3%). This implies your impressive efficiency gains for SLMs are entirely dependent on having access to dense, process-level rewards, which are notoriously expensive and difficult to create for new tasks. How do you reconcile this? Doesn't this limit the practical utility of your "SLM-First" approach to only tasks where we can already afford to create dense process rewards?**
>
> Thanks for highlighting the limitation around access to dense, process level reward as pre-requisite. We are primarily showing that in environments where process rewards are clear and can be reliably evaluated, smaller models become a viable and far more compute-efficient alternative.  In settings where process rewards are difficult to define, larger models are still the preferred choice. Please refer to the answer to W1.
>
> > **Q4 (On 32B Variance): You note that the 32B model exhibited "higher variance", and the CIs in Table 1 confirm this (e.g., 44.3 ± 14.1). Why do you think the larger model was less stable during training? That seems counter-intuitive.**
>
> TRAVELPLANNER is a challenging environment where essential information is often missing, requiring the agent to adapt, recover, and explore alternative solution paths. Our hypothesis is that smaller models explore less and behave more deterministically, as their limited capacity concentrates probability mass on a narrow set of actions, even when those actions are suboptimal. Larger models, by contrast, can consider a broader set of strategies under uncertainty (for example, switching transportation modes or hotel choices when details are missing). We therefore attribute the higher variance of the 32B model to its greater exploratory behavior during RL, amplified by the inherently uncertain nature of the task.
>
> We reflected that in the evaluation section:
> “Although the 32B model attained a slightly higher peak accuracy (56.9% vs. 56.4%), this difference was not statistically significant and was accompanied by higher variance. **We hypothesize that this variance stems from differences in exploration dynamics. TRAVELPLANNER is a challenging environment where essential information is often missing, requiring recovery and alternative solution paths. Smaller models tend to explore less and behave more deterministically, concentrating probability mass on a narrow set of actions. Larger models, by contrast, can consider a broader range of strategies under uncertainty (for example, switching transportation modes or hotel choices), leading to higher variability across rollouts. This exploratory flexibility likely contributes to the increased variance observed in the 32B runs**.
> ”

---

> ### Author Response · Authors · 2025-11-21
> **Part-4**
>
> > **Q5 (On the Solver Gap): You correctly position your 56.9% as SOTA for end-to-end agents. But you also note that the hybrid "solver" method (Hao et al., 2025) gets ~94-97% on this same benchmark. Do you believe that this massive ~37% gap can ever be closed by pure RL agent methods (e.g., with more data or better algorithms), or does this gap suggest that for hard-constraint tasks, hybrid neuro-symbolic methods will always be superior?**
>
> That’s an excellent point and an important discussion. We believe several promising directions could help close this gap. As detailed in Appendix C, and shown in Figure A.2, valid-cost violations account for the majority of remaining failures even in the later stages of training. This suggests that the current RL algorithm struggles to satisfy hard constraints in a single attempt. Recent work shows that giving the model a second chance—or multiple opportunities—to self-verify and self-correct can significantly improve both reasoning and verifiability (e.g., SRPO [1]: Reflection-Aware RL [2]; VL-Rethinker [3]). Incorporating similar reflection or correction mechanisms, combined with more targeted reward designs, may substantially narrow the gap between pure RL agents and hybrid solver approaches on constraint-heavy tasks.
>
> Orthogonal to the above, the model may also benefit from more intelligent sampling strategies — ranging from simply generating a larger number of rollouts (at higher computation cost) to increase the likelihood of successful trajectories, to more advanced approaches such as oversampling and then heuristic downsampling to retain only the traces that provide the most informative learning signals. Recent work such as rStar2-Agent: Agentic Reasoning Technical Report [4] and DAPO: An Open-Source LLM Reinforcement Learning System at Scale [5] demonstrates that strategic sampling can significantly improve training efficiency and stability.
>
> We expanded the discussion section with this information.
>
> [1] Wan, Zhongwei, et al. "Srpo: Enhancing multimodal llm reasoning via reflection-aware reinforcement learning." arXiv preprint arXiv:2506.01713, 2025.
>
> [2] Deng, Hexuan, et al. "REA-RL: Reflection-Aware Online Reinforcement Learning for Efficient Large Reasoning Models." arXiv preprint arXiv:2505.19862, 2025.
>
> [3] Wang, Haozhe, et al. "VL-rethinker: Incentivizing self-reflection of vision-language models with reinforcement learning." arXiv preprint arXiv:2504.08837, 2025.
>
> [4] Shang, Ning, et al. "rstar2-agent: Agentic reasoning technical report." arXiv preprint arXiv:2508.20722, 2025.
>
> [5] Yu, Qiying, et al. "DAPO: An open-source LLM reinforcement learning system at scale." arXiv preprint arXiv:2503.14476, 2025.
>
> > **Q6 (On Disabling "Thinking"): You mention you disabled the Qwen3 <think> token due to memory/context constraints. Since this "thinking" is a form of internal monologue (like Chain-of-Thought), is it possible this crippled the 32B model's ability to use the process-level rewards? In other words, perhaps the 32B model needed that "thinking" scratchpad to properly internalize the dense reward signals, and by removing it, you inadvertently made it look less responsive to reward shaping than it truly is?**
>
> We appreciate the reviewer’s perspective. We did experiment with Qwen3 `<think>` mode, but performance was significantly worse without it. The underlying reason is that `<think>` tokens consume the context window very inefficiently. Even with a 30.5k budget, the internal monologue rapidly fills the buffer, leading to frequent truncation of tool interactions. As shown in the new **Appendix I**, early-training trajectories of 32B model were clipped, meaning the agent receives reward only for short, partial rollouts. This biases learning toward short plans and limits performance on long-horizon queries. Disabling `<think>` preserves context for tool calls, yielding more stable rollouts and more effective learning under dense rewards. Improving the efficiency of thinking traces is an interesting direction but is beyond the scope of this work.

---

### Official Review · Reviewer_1Vmv · 2025-11-09

**Soundness:** 3
**Presentation:** 3
**Contribution:** 3
**Rating:** 6
**Confidence:** 3

**Summary:**

PLANNER-R1 studies RL for tool-augmented planning with LLMs on the TRAVELPLANNER benchmark. The authors show that dense reward shaping enables stable and efficient RL training, especially for smaller models like Qwen3-8B, which become competitive with much larger models at a fraction of the compute. They analyze hallucinations, tool sequencing, and constraint violations, and evaluate generalization across several out-of-domain tasks. The final PLANNER-R1 models substantially outperform GPT-5 in final-pass accuracy and do not exhibit overfitting. The paper argues that careful reward design is key for scalable, sample-efficient agentic RL and effective tool-use planning.

**Strengths:**

1. PLANNER-R1 achieves a notably higher final-pass rate compared to GPT-5, reaching 56.9 percent versus 21.2 percent. This is a considerable improvement in a challenging tool-augmented planning setting. The magnitude of the gain highlights the effectiveness of the proposed RL approach and positions the method as a meaningful advancement for agentic LLM planning.
2. The paper provides detailed examinations of common failure modes, including hallucinations, tool-use sequencing errors, and violations of hard or commonsense constraints, and tracks how these evolve throughout RL training. These analyses help clarify not only why the RL signal improves policy behavior but also how different model scales respond during learning. This level of insight strengthens the paper by making the empirical results more interpretable and actionable.
3. The evaluation on external benchmarks demonstrates that RL fine-tuning does not lead to overfitting to TRAVELPLANNER’s schema or task structure. Instead, the fine-tuned models often match or exceed their pretrained baselines across multiple out-of-domain tasks. This contrasts with conclusions from earlier RLVR-style work, where RL often harmed transferability. The results here suggest that, with well-designed rewards, RL can generalize more broadly and support effective planning and tool-use behaviors beyond the training environment, which is an encouraging direction for future research.

**Weaknesses:**

1. The paper reports several intriguing qualitative observations, such as the 8B model being highly sensitive to sparse rewards while the 32B model remains stable, and the 8B model achieving significantly better FLOPs efficiency. However, the authors stop at reporting these patterns without attempting to analyze or explain the underlying causes. These trends are not intuitively obvious, and without further discussion it is unclear whether they arise from architectural properties, optimization dynamics, or idiosyncrasies of the TRAVELPLANNER environment. This makes it difficult for readers to assess whether these findings would generalize to other domains or remain benchmark-specific.
2. Reward shaping is central to the paper’s claims, yet the discussion of the reward function is limited. The paper would benefit from ablations isolating the contribution of each reward component, or at least a more detailed discussion of how the shaping coefficients were chosen and tuned. Since reward design is critical to reproducing the reported improvements, more transparency would provide valuable guidance for practitioners looking to transfer these ideas to new tasks or environments.
3. igure 4 is difficult to interpret. The 32B model appears to exhibit higher average failure rates across several categories, yet achieves a higher final-pass rate. This seemingly contradictory behavior raises questions: which failure types meaningfully affect the final evaluation metric, and which do not? Are some hallucinations or constraint violations “benign” in the sense that they do not influence schema validity or final constraints? The paper should clarify how these failure metrics relate to task success, otherwise the figure may confuse readers and undermine the interpretability of the qualitative analysis.
4. Although the paper lists the RL benchmark formulation as a key contribution, it offers limited detail beyond high-level framing. The work would benefit from clearer articulation how the environment proposed by the paper differ from the original TRAVELPLANNER setup.

**Questions:**

1. What underlying factors explain why the 8B model is highly sensitive to sparse rewards while the 32B model remains stable, and why the smaller model achieves superior FLOPs efficiency? Are these effects specific to TRAVELPLANNER, or should we expect similar patterns in other domains?
2. How were the reward coefficients selected and tuned, and what is the contribution of each reward component? Can the authors provide ablations or design principles that would help practitioners adapt the reward function to new environments?
3. Why does the 32B model show higher average failure rates yet obtain higher final-pass performance? Which types of failures meaningfully impact the final metric, and which ones are effectively ignored or compensated for during evaluation?
4. In what specific ways does the proposed RL benchmark formulation differ from the original TRAVELPLANNER environment, and what elements of it constitute the claimed contribution?

---

> ### Author Response · Authors · 2025-11-21
> **Part-1**
>
> We thank the reviewer for the insightful comments and appreciate their recognition of PLANNER-R1’s substantial final-pass gains, our detailed analysis of evolving failure modes, and the strong out-of-domain generalization. Below we address your weaknesses and questions:
>
> > **W1: What underlying factors explain why the 8B model is highly sensitive to sparse rewards while the 32B model remains stable, and why the smaller model achieves superior FLOPs efficiency? Are these effects specific to TRAVELPLANNER, or should we expect similar patterns in other domains?**
>
> **Reward Model Sensitivity**:
>
> Sparse rewards yield slower convergence speed for policy optimization compared to dense rewards, given the same underlying model (we provide a theoretical proof for this claim in appendix G). Smaller models lack the capacity to discover successful final-pass trajectories under sparse outcome rewards. In the Stage-3 setting, where a positive reward is given only when all hard constraints are satisfied, the 8B model often fails to produce even a single correct trajectory early in training. This leads to near-zero advantage values across the rollout batch, preventing effective learning within a fixed compute budget. The 32B model preserving more exploratory power can be relatively robust given sparse rewards.
>
> **FLOPS Efficiency**:
>
> In the scenario of dense rewards with better convergence speed, the smaller model (8B) requires less computation to update its policy due to its smaller parameter size, resulting in higher FLOPs efficiency. These effects are not specific to TRAVELPLANNER, we should expect similar patterns in other domains as well.
>
> **Generality of Results**: (Same response to R3-W1)
>
> We appreciate the reviewer’s concern. The goal of this paper is not to introduce a new reward-shaping algorithm, but to provide the insight that **small models can be strong and far more compute-efficient agentic RL learners when dense, well-structured process rewards are available**. With this framing, we agree that evaluating additional domains would further strengthen generality. However, our results are consistent with a broader empirical trend: **7–8B models approach the performance of 30–32B models** in simpler domains. Examples include:
>
> | Dataset | Small Model | Large Model | Small Perf. | Large Perf. | Paper |
> |--------|-------------|-------------|-------------|-------------|--------|
> | HumanEval (pass@1) | Qwen2.5-Coder-7B-Instruct | Qwen2.5-Coder-32B-Instruct | 88.4% | 92.7% | arXiv:2409.12186 |
> | GSM8K (CoT), 8-shot | Phi-3 Small (7B) | Phi-3 Medium (14B) | 89.6% | 91% | arXiv:2404.14219 |
>
> Beyond this general trend, **TRAVELPLANNER is already a highly challenging agentic environment**, not a toy domain. Even GPT-5 (high reasoning effort) achieves only about 20% final-pass accuracy. The agent must interpret complex travel intents, autonomously call APIs, handle missing information, and satisfy dozens of constraints across trajectories that often exceed 30.5k tokens. These properties reflect real-world long-horizon planning much more closely than synthetic benchmarks.
>
> To ensure robustness and reproducibility, we ran **five independent seeds per configuration**, all using identical scripts, random seeds, curriculum schedules, and reward setups. All runs were executed on **two H200 nodes (16 GPUs)** with full logging and fixed pipelines to eliminate infrastructural variance. Running five seeds for 500 steps for both 8B and 32B models cost approximately **$2K** and **$7K** respectively, bringing the minimum cost of the study to **$67K** across all configurations. Additional exploratory sweeps roughly doubled this footprint. Given the importance of long-horizon planning tasks for agentic RL, we view this investment as both meaningful and necessary.

---

> ### Author Response · Authors · 2025-11-21
> **Part-2**
>
> > **W2: How were the reward coefficients selected and tuned, and what is the contribution of each reward component? Can the authors provide ablations or design principles that would help practitioners adapt the reward function to new environments?**
>
> We use binary coefficients (0 or 1) to include or exclude each reward component, and we do not tune these coefficients beyond this structural choice. Each component corresponds to a specific part of the task decomposition: micro constraints provide fine-grained partial credit for satisfying portions of constraints within a category, macro constraints provide categorical rewards when an entire category is met, and the final-pass reward reflects satisfaction of all categories. Appendix F in our revision now enumerates all subconstraints within the two categories: commonsense and hard constraints. Our reward design follows a principled densification path: starting from the sparse Stage 3 reward, we add macro structure (Stage 2) and then micro structure (Stage 1), ensuring that each component is a valid refinement of the overall task objective. Table 1 acts as an ablation across these stages and shows that increasing reward density significantly improves learning for smaller models while not impacting larger models. We did not further isolate individual component contributions nor did we investigate various reward designs. As Reviewer 2 and 4 highlighted, there are encouraging body of work for creating such dense rewards such as ORSO and ToolRL.
>
> > **W3: (Figure 4) Why does the 32B model show higher average failure rates yet obtain higher final-pass performance? Which types of failures meaningfully impact the final metric, and which ones are effectively ignored or compensated for during evaluation?**
>
> Thanks for the careful reading of the plot. To receive a final-pass reward, all evaluation dimensions must be satisfied, so none of the failure types are ignored or compensated for in the scoring pipeline. The apparent contradiction arises because the figure reports only the top five failure categories across the test set. While the 32B model shows higher failure rates in that specific subset, it fails less frequently in the remaining categories not shown in the plot. When averaged over the complete distribution of error types, the 32B model therefore achieves a higher overall final-pass rate despite higher error incidence within the plotted subset. Figure A.1 in the appendix displays the evolution of all six subelements of the reward function for **validation data**, showing that the 32B model performs better on the full set of constraints in most cases, yet the final pass rate difference was not statistically significantly different.
>
> > **W4: In what specific ways does the proposed RL benchmark formulation differ from the original TRAVELPLANNER environment, and what elements of it constitute the claimed contribution?**
>
> As noted in lines 154–157, our benchmark differs from the original TRAVELPLANNER environment in two modeling changes and one system change. First, we added a `calculator` API that enables explicit numeric reasoning. Second, we removed the lightweight semantic memory so that tool responses are written directly into the context and all tokens, including tool calls and responses, are appended chronologically, with the final itinerary produced in a standardized `<answer>...</answer>` format; this yields a more generic and easily reusable transition structure. Third, we introduced system-level optimizations that reduce runtime and memory usage by 20% percent, enabling efficient large-scale experimentation.
>
> > **Q1: What underlying factors explain why the 8B model is highly sensitive to sparse rewards while the 32B model remains stable, and why the smaller model achieves superior FLOPs efficiency? Are these effects specific to TRAVELPLANNER, or should we expect similar patterns in other domains?**
>
> Please refer to the answer to W1.
>
> > **Q2: How were the reward coefficients selected and tuned, and what is the contribution of each reward component? Can the authors provide ablations or design principles that would help practitioners adapt the reward function to new environments?**
>
> Please refer to the answer to W2.
>
> > **Q3: Why does the 32B model show higher average failure rates yet obtain higher final-pass performance? Which types of failures meaningfully impact the final metric, and which ones are effectively ignored or compensated for during evaluation?**
>
> Please refer to the answer to W3.
>
>
> > **Q4: In what specific ways does the proposed RL benchmark formulation differ from the original TRAVELPLANNER environment, and what elements of it constitute the claimed contribution?**
>
> Please refer to the answer to W4.

---

> > ### Comment · Reviewer_1Vmv · 2025-11-25
> >
> > Thank you for the detailed and thoughtful responses. They fully address all of my concerns. With these clarifications, I am satisfied with the soundness and contribution of the work and will maintain my recommendation to accept the paper.

---

### Comment · Area_Chair_8HHK · 2025-11-26
**Reminder of Discussions**

Hi Author and Reviewers,

Please remember that the manuscript is editable during the rebuttal. And it is appreciated if all of you can engage in the discussion.

Best,
AC

---

> ### Author Response · Authors · 2025-11-26
>
> Thank you for the reminder. We have responded to all reviewer comments, strengthened the theoretical grounding and analysis in the appendix, and updated the main sections in line with the reviewers’ suggestions. The revised parts are clearly indicated in our responses. We greatly appreciate the constructive feedback and the positive assessment of our contributions, and we look forward to further discussion.

---

### Meta-Review · Area_Chair_qxbr · 2025-12-30

**Summary:**

This paper investigates agentic reinforcement learning for large language models (LLMs) on the TravelPlanner benchmark, focusing on enabling small models (8B) to achieve competitive performance through multi-stage dense reward shaping. Reviewers consistently acknowledged the strong empirical results, robust experimental setup with multiple seeds and confidence intervals, and practical relevance of efficient post-training for lightweight models. The authors’ responses addressed several concerns: they added theoretical justification showing policy consistency and faster convergence under dense rewards, clarified reward definitions and curriculum schedules, and explained the behavior of small versus large models and the limited impact of curriculum learning. However, the work remains primarily empirical, evaluated on a **single domain** with only two model sizes, limiting its generalizability and novelty. While the study provides valuable practical guidance, it does not introduce fundamentally new methods or broadly applicable theoretical insights.

Overall, despite the thorough rebuttal and improvements, the paper is not sufficiently generalizable or methodologically novel for acceptance, and I recommend rejection.

**Reviewer Concerns:**

**Theoretical Justification**

No guarantee that shaped rewards preserve optimal policy; lack of theoretical understanding of convergence or asymptotic behavior (auEt, zmo9). The authors have added theoretical proofs to the appendix to clarify this.

**Generalization and Scope**

Experiments are limited to TravelPlanner, raising questions about applicability to other tasks; only two model sizes were evaluated (auEt, zmo9, 1Vmv). This is the major concern, still outstanding. Without additional tasks, it is hard to verify that the findings claimed in the paper hold.

**Reviewer Scores:**

I do not think Reviewer auEt will change the score, based on the analysis above.

---

### Decision · Program_Chairs · 2026-01-26

Reject